# CCDC66 regulation of cytoskeleton and cilia stability is important for signaling and epithelial organization

Jovana Deretic[1], Seyma Cengiz-Emek[1]☉, Ece Seyrek[1]☉, Elif Nur Firat-Karalar [1,2]*

1 Department of Molecular Biology and Genetics, Koç University, Istanbul, Turkey, 2 School of Medicine, Koç University, Istanbul, Turkey

☉ These authors contributed equally to this work.
* ekaralar@ku.edu.tr

## Abstract

The primary cilium is a conserved, microtubule-based organelle that transduces signaling pathways essential for development and homeostasis. It dynamically assembles and disassembles in response to intrinsic and extrinsic stimuli while maintaining remarkable structural stability and tightly regulated length. The mechanisms underlying this stability and length control are not well understood. Here, we characterized CCDC66, a microtubule-associated protein linked to ciliopathies, as an important regulator of cilium maintenance and disassembly in mouse epithelial cells. Live imaging revealed that cilia in CCDC66-depleted cells frequently fluctuate in length and exhibit increased cilium disassembly and ectocytosis. Phenotypic rescue experiments and in vitro assays showed that microtubule stabilization activity of CCDC66 is required for these functions. Temporal proximity mapping identified potential new regulators and molecular pathways involved in cilium disassembly. Further characterization revealed actin cytoskeleton and vesicular trafficking as additional mechanisms by which CCDC66 may mediate its ciliary functions. Finally, depletion of CCDC66 compromised Hedgehog and Wnt pathway activation and disrupted epithelial cell organization and polarity in two- and three-dimensional cultures. Collectively, we showed that CCDC66 regulates both ciliary and non-ciliary processes through diverse mechanisms involving microtubules, actin, and vesicular trafficking, providing insights into the pathologies associated with CCDC66.

## Introduction

The primary cilium is an evolutionary conserved, microtubule-based cellular projection that transduces signaling pathways essential for development and homeostasis, including Hedgehog and Wnt signaling [1,2]. It is composed of a microtubule-based axoneme surrounded by a ciliary membrane. The nine, radially arranged microtubule

**Data availability statement:** All relevant data are within the paper and its Supporting Information files.

**Funding:** This work was supported by European Research Council Starting Grant 679140 to ENF-K and 101078097 to ENF-K (https://erc.europa.eu/starting-grants), European Molecular Biology Organization Installation Grant 3622 to ENF-K (https://www.embo.org/funding-awards/installation-grants), European Molecular Biology Organization Young Investigator Award to ENF-K (https://www.embo.org/funding-awards/young-investigators), The Scientific and Technological Research Institution of Turkey BIDEB Grant 120C148 to ENF-K (https://tubitak.gov.tr/tr/burslar/doktora-sonrasi/arastirma-burs-programlar-i/2247-ulusal-lider-arastirmacilar-programi), Marie Sklodowska-Curie Grant 896644 to JD (https://marie-sklodowska-curie-actions.ec.europa.eu/actions/postdoctoral-fellowships) and The Scientific and Technological Research Institution of Turkey ARDEB Grant 120Z179 to JD (https://tubitak.gov.tr/tr/destekler/akademik/ulusal-destek-programlari/1001-bilimsel-ve-te-knolojik-arastirma-projelerini-destekleme-pro-grami) and TUBITAK BIDEB 2211 fellowship TBTK-0065-7203 to SC (https://tubitak.gov.tr/tr/burslar/lisansustu/egitim-burs-program-lari/2211-yurt-ici-doktora-burs-programlari). The funders had no role in study design, data collection and analysis, decision to publish, or preparation of the manuscript.

**Competing interests:** The authors have declared that no competing interests exist.

**Abbreviations :** EV, extracellular vesicles; IFT, intraflagellar transport; MAP, microtubule-associated proteins; PTM, posttranslational modifications; shRNA, short hairpin RNA.

doublets of the axoneme are exceptionally stable, mechanically resistant and highly modified by posttranslational modifications (PTMs) [3–6]. The axoneme supports the shape and structure of the cilium and acts as a track for ciliary transport complexes [7,8]. The ciliary membrane is continuous with the plasma membrane, but distinct in its lipid and protein composition. Sensory functions of the cilium require tight spatio-temporal control of its assembly kinetics, length, stability, structure and composition. Deregulation of these processes causes various human diseases including cancer and the multisystem pathologies of the eye, kidney, skeleton, brain and other organs, collectively named "ciliopathies" [9,10]. Defining the mechanisms by which a functional cilium is assembled, maintained and disassembled in response to different signals and across different cell types is essential to uncover the molecular defects underpinning these diseases.

Primary cilium assembly, maintenance and disassembly are highly regulated processes governed by intrinsic and extrinsic stimuli such as cell cycle cues [8,11]. In most cells, primary cilium assembles during differentiation and quiescence. The assembly process involves the maturation of the mother centriole into a basal body, the elongation of the axoneme from the basal body by Intraflagellar Transport (IFT)-mediated transport of ciliary components and the formation of the ciliary membrane and the ciliary gate [11]. Primary cilium forms via extracellular and intracellular pathways, classified based on whether cilium growth is initiated at the cell surface or within the cytoplasm, respectively [12,13]. The extracellular pathway is commonly observed in epithelial cells of the kidney and lung, whereas the intracellular pathway is typical in fibroblasts and retinal epithelial cells.

Once a cilium is formed, it maintains its length, integrity and structure despite the continuous turnover of lipids and proteins, mediated by mechanisms such as the IFT machinery, BBSome complex and vesicular trafficking [4,14]. Cilium shortening, over-elongation and instability have been reported in ciliopathies, highlighting the physiological significance of maintaining proper cilium length and stability for its functions [10,15–18]. To date, a variety of mechanisms regulating cilium length and stability have been uncovered through functional screens and targeted molecular and cellular studies. These studies have identified proteins involved in diverse cellular components and functions, including cell cycle regulation, organization of the actin and microtubule cytoskeleton, lipid metabolism, vesicle trafficking, and transcription [19–29]. Within this context, microtubule-associated proteins (MAPs) have been recognized as critical regulators of these ciliary properties by acting on both axonemal and cytoplasmic microtubules [7,30]. The best characterized MAPs for axoneme length control are the molecular motors that cooperate with IFT machinery [31]. However, the roles of non-IFT ciliary MAPs remain less understood. Further identification and characterization of these MAPs are necessary to elucidate the mechanisms by which axonemal microtubules are elongated from the basal body to the correct length and how their length and stability are maintained.

Upon mitotic entry, the primary cilium disassembles primarily via two pathways: resorption and deciliation. Resorption involves the depolymerization of the axoneme and incorporation of ciliary components into the cell. This process is regulated by the

HEF1-Aurora A kinase (AURKA)-histone deacetylase 6 (HDAC6) pathway, which leads to the deacetylation and subsequent destabilization of axonemal microtubules [32,33]. Ciliary resorption is typically preceded by the release of vesicles from the distal end of the cilium, a process known as ectocytosis or ciliary decapitation, which is driven by intraciliary actin polymerization. The second pathway, deciliation, involves severing the axoneme near its base by the microtubule-severing enzyme katanin, as reported in mouse kidney epithelial cells [8,34,35]. Despite advances in understanding cilium disassembly mechanisms, the specific roles of different cytoskeletal elements and their associated proteins, such as the non-IFT MAPs, remain less understood.

CCDC66 is a MAP that plays important roles in primary cilium assembly, ciliary signaling and the progression of mitosis and cytokinesis [36–38]. It exhibits a dynamic localization profile, ranging from localization to the ciliary axoneme, centrosome and centriolar satellites in quiescent cells, to the mitotic spindle, the central spindle and the midbody in dividing cells [36,38,39]. Genetic studies in dogs and mice have linked early frameshift mutations and deletions of CCDC66 to retinal degeneration and olfactory deficits [40–43]. Moreover, CCDC66 is part of a ciliary tip module that includes other MAPs such as CEP104, CSPP1, TOGARAM1 and ARMC9, which are mutated in the Joubert syndrome [44,45]. In vitro reconstitution experiments suggest that CCDC66 directly regulates microtubule organization, polymerization, stabilization, and actin-microtubule crosstalk [37,38,46,47]. Although these studies identify CCDC66 as a key regulator of cytoskeletal processes and intracellular cilium assembly, its roles in extracellular ciliogenesis pathways remain unknown. Additionally, its involvement in other stages of the cilium life cycle, particularly maintenance and disassembly, has yet to be explored.

Here, we used localization, proximity mapping and loss-of-function experiments to define the functions and mechanisms of CCDC66 in mouse inner medullary collecting duct (IMCD3) cells. We chose IMCD3 cells because they respond to Hedgehog signaling, form epithelial spheroids in three-dimensional matrices, and ciliate via the extracellular pathway, whereas RPE1 cells use the intracellular pathway. We discovered that CCDC66 is required for maintaining cilium length and stability and controls cilium disassembly. Functional assays, temporal proximity mapping during cilium disassembly and in vitro microtubule stabilization assays collectively indicate that CCDC66 functions as a microtubule-stabilizing factor while also regulating the actin cytoskeleton and vesicular trafficking. Ciliary and non-ciliary functions of CCDC66 is required for epithelial cell organization and polarity in two-dimensional and three-dimensional cultures. Overall, our results identify CCDC66 as a regulator of cilium homeostasis, elucidate its role in cellular signaling and tissue organization and provide insight into pathologies linked to CCDC66.

## Results

### CCDC66 localizes to the ciliary axoneme and regulates cilium length and stability

To examine the endogenous localization of CCDC66 in IMCD3 cells, we generated a custom antibody and stained ciliated IMCD3 cells with it, along with antibodies against axonemal microtubules (acetylated tubulin) and ciliary membrane (ARL13b) (Fig 1A). For nanoscale mapping of CCDC66 localization at the primary cilium, we analyzed its localization using Structured Illumination Microscopy (SIM) and Ultrastructural Expansion Microscopy (U-ExM). Both imaging approaches showed that CCDC66 co-localized with acetylated tubulin at the axonemal microtubules and basal body (Fig 1A). CCDC66 was heterogeneously distributed along the ciliary axoneme in a punctate pattern (Fig 1A). However, we did not observe a prevalent ciliary tip pool with endogenous staining [39]. These results confirm endogenous localization of CCDC66 to the ciliary axoneme in mouse cells.

We performed loss-of-function experiments to determine the role of CCDC66 during cilium assembly and maintenance in IMCD3 cells. First, we depleted CCDC66 from IMCD3 cells by lentivirus-mediated short hairpin RNA (shRNA) treatment. Immunofluorescence, immunoblotting and quantitative PCR experiments confirmed efficient depletion of CCDC66 with varying levels in IMCD3 cells 6 days after transduction (S1A–S1C Fig). The loss of CCDC66 ciliary and midbody signals in cells transduced with Ccdc66 shRNA compared to control cells validated the specificity of the rat CCDC66 antibody (S1A Fig) [36]. We then examined the effects of CCDC66 depletion on cilium assembly and length by staining cells for

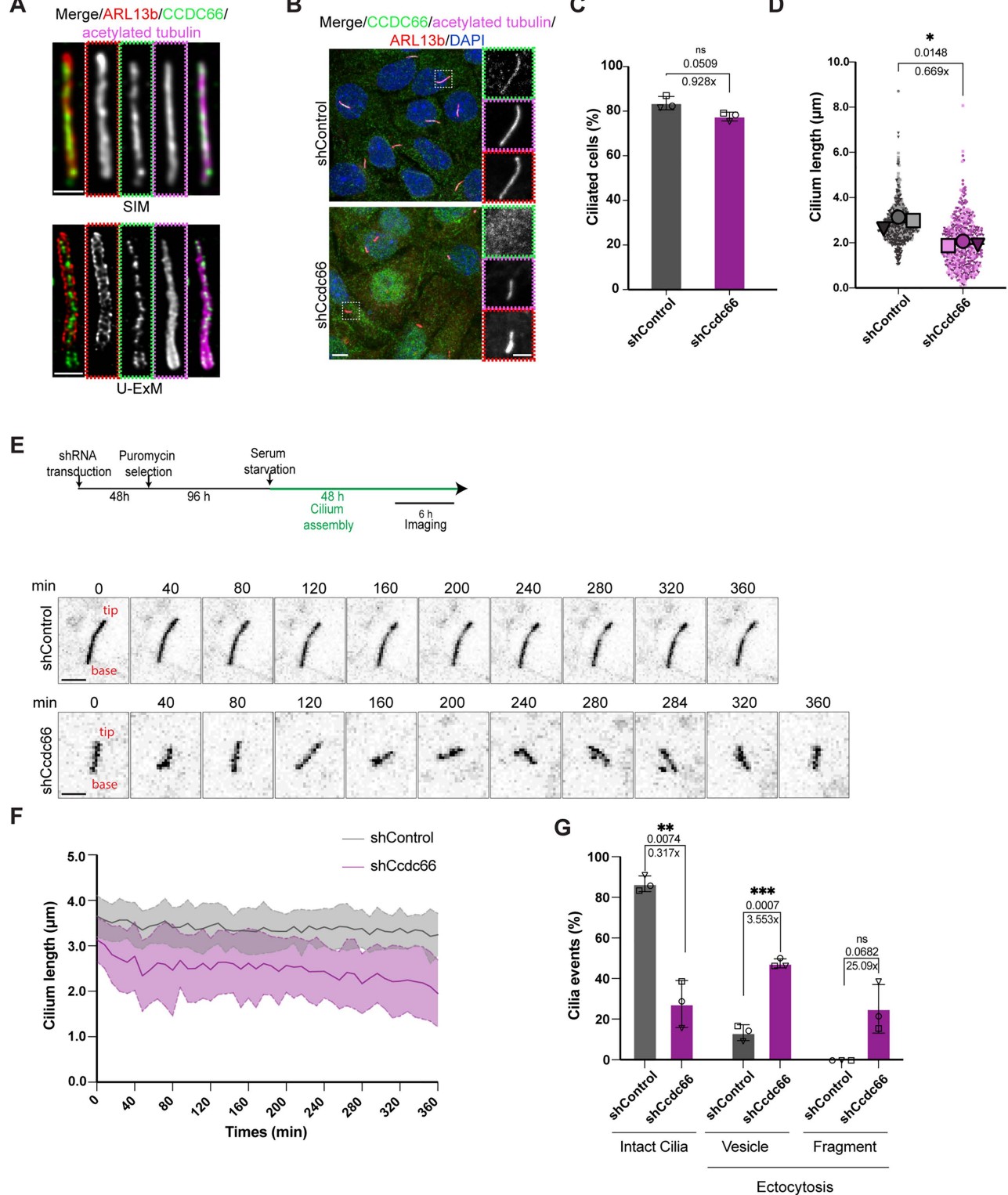

**Fig 1. CCDC66 localizes to the axoneme and regulates primary cilium stability but not cilia formation in IMCD3 cells. (A)** CCDC66 localizes to the axoneme of the primary cilium. Endogenous staining of CCDC66 in ciliated IMCD3 cells fixed with 4% PFA was performed using homemade antibody raised against the mouse CCDC66 protein. Cells were co-stained with anti-acetylated-tubulin, anti-ARL13b and DAPI. The top panel shows

images obtained using structured illumination microscopy (SIM). The bottom panel shows confocal microscopy images of samples processed by ultra-structure expansion microscopy (U-ExM). Scale bars SIM: 1 µm, Scale bars U-ExM: 222 nm. **(B–D)** Effects of CCDC66 depletion on cilia formation and length. IMCD3 cells transduced and stably expressing either control or shCcdc66 were grown on glass coverslips, fixed with 4% PFA after 48 h of serum starvation and imaged with confocal microscopy. Scale bar: 5 µm. Insets show 3× magnifications of the cilia, Scale bar: 2 µm **(C)** Quantification of percentage of cells with cilia from **(B)**. Data represents the mean±SD of 3 independent experiments. $n > 600$ cells for control and >400 cells for CCDC66 depletion. Mean cilia percentage is 83.62% for shControl and 77.60% for shCcdc66. (Welch's $t$ test, ns: not significant) **(D)** Quantification of cilia length in 3D, according to ARL13b signal (co-localizing with acetylated-tubulin) from **(B)**. The super plot represents the mean±SEM of 3 independent experiments superimposed onto a scatter plot of individual experimental values. The first experimental replicate is shown as a darker circle, the second as a lighter-colored square, third as the darkest triangle. Individual values are represented in lighter shades than the corresponding averages. $n > 600$ cells for control and >400 cells for CCDC66 depletion. The mean cilia length in CCDC66-depleted cells decreased to 0.67-fold compared to the mean control length. (Welch's $t$ test, *$p = 0.0148$). **(E-G)** CCDC66 depletion leads to frequent length fluctuations and enhanced vesicle ectocytosis and cilia fragmentation in steady-state cilia. **(E)** The IMCD3::SSTR3-GFP cells transduced with either control or Ccdc66 shRNA virus were grown in FluoroDish and serum starved for 48 h, then imaged with confocal microscopy with 63× objective. Images were acquired every 4 min for 6 h. Still images are inverted to emphasize cilia better. Scale bar: 3 µm. **(F)** Ciliary kinetics are presented as raw length curves, measured from SSTR3-GFP fluorescence in three independent experiments, represented as the mean±SD. $n = 30$ cilia for both shControl and shCcdc66 conditions. **(G)** Quantification of ciliary ectocytosis events in **(E)**. Ciliary vesicle ectocytosis or fragments were categorized based on the size of the released EVs, vesicle < 500 nm and fragment > 500 nm, and plotted as a bar plot. ($p$ values of Welch's $t$ test, **$p = 0.0074$, ***$p = 0.0007$, ns = not significant). Statistical analysis is performed on the means of 3 experimental replicates. The data underlying the graphs shown in the figure can be found in S1 Data. CCDC66, coiled-coil domain-containing protein 66; IMCD3, inner medullary collecting duct cell line; PFA, paraformaldehyde; ARL13b, ADP-ribosylation factor-like protein 13B; DAPI, 4′,6-diamidino-2-phenylindole; U-ExM, Ultra-structure expansion microscopy; shRNA, short hairpin RNA; SSTR3, Somatostatin receptor type 3; GFP, green fluorescent protein; EV, extracellular vesicle; SEM, standard error of mean; SD, standard deviation.

acetylated tubulin and ARL13b. The percentage of ciliated CCDC66-depleted cells was comparable to that in control cells 48 h post serum starvation (Fig 1B and 1C). However, the cilia that formed in CCDC66-depleted cells were significantly shorter and exhibited length variability (Figs 1D and S1D). These results show that CCDC66 is required for cilia elongation but not initiation of cilium assembly in IMCD3 cells.

We next analyzed the dynamics, stability and length of steady state cilia by spatiotemporally measuring these ciliary properties. To this end, we performed 3D confocal live imaging of ciliated control and CCDC66-depleted IMCD3 cells stably expressing the GFP fusion of the ciliary transmembrane protein SSTR3 (GFP-SSTR3) (Fig 1E). While cilium length remained relatively constant in control cells, we observed significant variations in length among CCDC66-depleted cells, suggesting defects in cilium stability (Figs 1E, 1F, and S1E). Since tubulin PTMs contribute to maintaining cilium stability and length, we quantified the ciliary concentration of polyglutamylated and acetylated tubulin [6]. However, we found no significant differences between control and CCDC66-depleted cells (S1F Fig). This result indicates that the observed cilium instability defect is not due to defects in axonemal tubulin modifications.

Ectocytosis, a process by which vesicles are shed from the cilium, regulates ciliary length, maintains ciliary stability and promotes cilium disassembly [23,48,49]. To investigate whether it contributes to the ciliary defects associated with CCDC66 depletion, we quantified ectocytosis events using live-imaging videos. Our analysis showed that the cilia in CCDC66-depleted cells frequently exhibited two different types of ectocytosis events: (1) vesicle release from the distal tip of the cilium, hereafter referred to as vesicle, and (2) breakage of ciliary fragments longer than 500 nm, hereafter referred to as fragment. About 66.5% of the cilia observed in CCDC66-depleted cells underwent vesicle and fragment release during the imaging period, compared to less than 15.5% in control cells (Fig 1G). Additionally, cilia in CCDC66-depleted cells showed an increase in the frequency of ectocytosis events. Collectively, these experiments show that CCDC66 depletion leads to fluctuations in cilium length with increased instability, highlighting its role in maintaining the stability of the steady-state cilia.

## CCDC66 regulates primary cilium stability by acting through microtubules and actin

Since CCDC66 is a MAP, we hypothesized that it might regulate cilium stability by acting as a microtubule-stabilizing factor. To test this, we first examined whether CCDC66 overexpression enhances stability of cilia by assessing their

resistance to microtubule depolymerization in control and CCDC66-expressing cells. Previous studies showed that high doses of nocodazole result in shortened cilia and increased cilium instability [19,50]. In our experiments, we used previously described IMCD3 cells that stably express Flag-miniTurbo and Flag-miniTurbo-CCDC66, (hereafter miniTurbo and miniTurbo-CCDC66) [38]. We treated these cells with 400 ng/ml of nocodazole for 2 h to induce microtubule depolymerization and challenge axonemal integrity (Fig 2A). We then measured the ciliary length difference between cells treated with vehicle control or nocodazole. In CCDC66-expressing cells, the cilium length after nocodazole treatment was 88.8% of its starting length, compared to 43.4% in control cells (Fig 2B). These results suggest that cilia in CCDC66-expressing cells are more stable than those in control cells.

CCDC66 promotes microtubule bundling in vitro, suggesting that it may stabilize microtubules by crosslinking them [36]. To directly examine the role of CCDC66 in microtubule stabilization, we performed in vitro microtubule stability experiments using MBP-mNeonGreen-CCDC66, which was expressed and purified from insect cells as described previously [36]. MBP-mNeonGreen, which does not bind to microtubules and the first two tubulin-binding TOG domains of TOGARAM1 were used as controls [47,51]. Specifically, polymerized microtubules were diluted 10-fold in prewarmed buffer lacking GTP and glycerol, with and without the indicated proteins, and the reaction was imaged in a flow chamber (Fig 2C). While microtubules disassembled in the presence of MBP-mNeonGreen and MBP-TOGARAM1$_{TOG1TOG2}$, microtubule bundles induced by MBG-mNeonGreen-CCDC66 persisted after dilution (Fig 2C). These results show that CCDC66 stabilizes microtubules against dilution-induced depolymerization.

In addition to microtubules, intraciliary actin was shown to be required for the stability of cilium, in part via regulating ciliary ectocytosis [23,48,52,53]. To assess whether regulation of steady-state cilium by CCDC66 involves actin, we examined whether the expression of CCDC66 affects cilia elongation induced by inhibition of actin polymerization [20,54]. For this purpose, we treated IMCD3 cells that stably express miniTurbo or miniTurbo-CCDC66 with a low dose of cytochalasin D (CytoD), an actin depolymerizing agent, and measured changes in cilium length. CCDC66 expression inhibited over-elongation of cilia upon CytoD treatment, supporting the role of actin in the CCDC66-mediated regulation of cilium length (Fig 2D and 2E).

Collectively, results from loss-of-function, overexpression, and in vitro experiments suggest that stabilization of microtubules and regulation of actin dynamics are mechanisms by which CCDC66 exerts its ciliary functions.

## CCDC66 depletion enhances cilium disassembly via destabilization of ciliary microtubules

The microtubule-stabilizing activity of CCDC66 may play a role during cilium disassembly, a process that involves destabilizing axonemal microtubules by the AURKA/HDAC6 pathway. To investigate this possibility, we measured the percentage of ciliated cells following cilium disassembly, which was induced by serum stimulation. IMCD3 cells infected with control and Ccdc66-targeting lentivirus were serum-starved for 48 h to induce cilium assembly. Subsequently, we quantified the percentage of ciliated cells over a 12 h serum stimulation time course (Fig 3A). Notably, there was a rapid and significantly greater decrease in the percentage of ciliated cells upon CCDC66 depletion as compared to control cells at both 6 and 12 h after serum stimulation (Fig 3A and 3B). Over 12 h, the percentage of ciliated cells in control cells decreased from 84.1% ± 2.5% to 73.8% ± 3.5%, while in CCDC66-depleted cells, it significantly dropped to 40.5% ± 10.2% (Fig 3B). Strikingly, majority of the CCDC66-depleted cells had ciliary stubs, identified by a short (<1 μm) cilium stained positive for both acetylated tubulin and ARL13b (Fig 3A).

To validate the specificity of the ciliary phenotypes, we performed phenotypic rescue experiments using IMCD3 cells stably expressing shRNA-resistant miniTurbo-CCDC66 [38]. In both control and CCDC66-depleted ciliated cells, miniTurbo-CCDC66 localized to the basal body, axoneme, ciliary tip and centriolar satellites (S2A Fig). Notably, the satellite pool of miniTurbo-CCDC66 became more prominent throughout the 12 h of serum stimulation (S2A Fig). Expression of miniTurbo-CCDC66 restored cilium length and reversed the enhanced cilium disassembly phenotypes in CCDC66-depleted cells to levels comparable to those to control cells, confirming phenotype specificity (S2A–S2C Fig). Using a

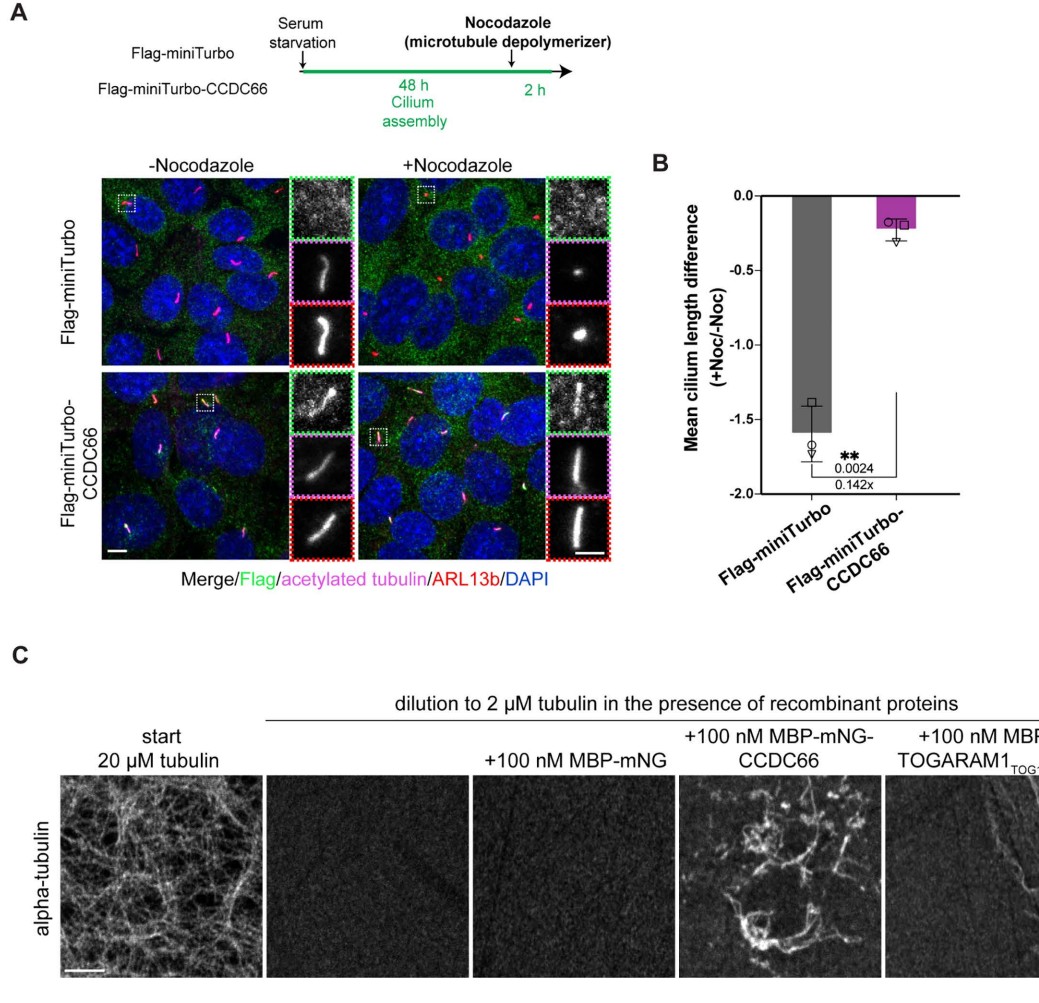

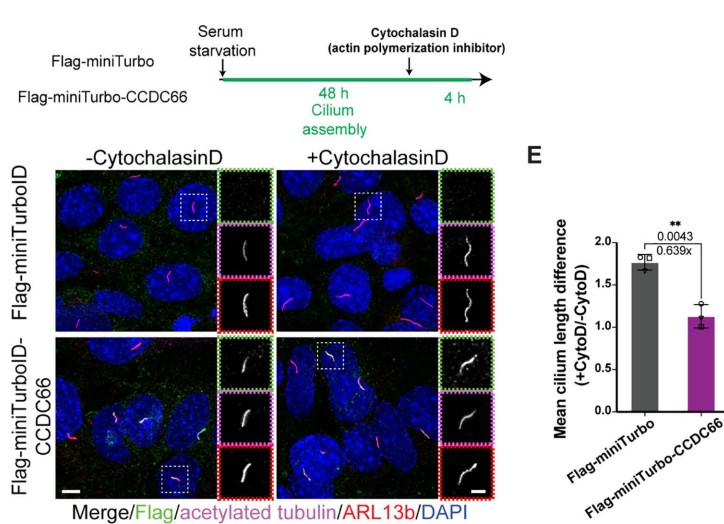

**Fig 2. CCDC66 stabilizes microtubules in cells and in vitro, and counteracts ciliary over-elongation induced by actin depolymerization. (A, B)** IMCD3::Flag-miniTurboID and IMCD3::Flag-miniTurboID- CCDC66 were seeded on coverslips. After 48 h serum starvation, the cells were treated with either DMSO or 400 ng/mL (1.33 μM) nocodazole for 2 h, which destabilizes cytoplasmic and axonemal MTs, fixed with 4% PFA and stained with

anti-Flag, anti-acetylated-tubulin, anti-ARL13b and DAPI. Scale bar: 5 μm. Insets show 3× magnifications of the cilia, Scale bar: 2 μm **(B)** Quantification of difference in average cilia length in +Noc relative to their starting length in −Noc treatment from **(A)**. Data represents mean ± SD of 3 independent experiments. $n > 50$ cilia for each cell line and treatment per experiment. The mean cilia length difference is −1.597 for miniTurbo control and −0.227 for miniTurbo-CCDC66. (p value of Welch's t test, \*\*\*p = 0.0024). **(C)** In vitro MT stabilization experiments were performed with MBP-mNeonGreen-CCDC66, MBP-mNeonGreen and MBP-TOGARAM1$_{TOG1TOG2}$. Tubulin (20 μM) was polymerized in BRB80 buffer, 1 mM DTT reducing agent, 1 mM GTP and 25% glycerol at 37°C for 30 min. Subsequently, microtubules were diluted to 2 μM tubulin concentration in the buffer without GTP and glycerol, absence, or presence of recombinant proteins at indicated concentrations. The reaction was incubated for 5 min at RT, followed by fixation with glutar-aldehyde, pelleting onto glass coverslips and additional staining with anti-tubulin antibody. Reactions were imaged using a confocal microscope. **(D)** IMCD3::Flag-miniTurboID and IMCD3::Flag-miniTurboID- CCDC66 were seeded on coverslips. After 48 h serum starvation, the cells were treated with either DMSO or 0.5 μM CytoD for 4 h, which prevents actin polymerization, fixed with 4% PFA and stained with anti-Flag, anti-acetylated-tubulin, anti-ARL13b and DAPI. Scale bar: 5 μm. Insets show 3× magnifications of the cilia, Scale bar: 2 μm **(E)** Quantification of difference in average cilia length in +CytoD relative to their starting length in −CytoD treatment from **(D)**. Data represents mean ± SD of 3 independent experiments. $n > 50$ cilia for each cell line and treatment per experiment. The mean cilia length difference is 1.769 for miniTurbo control and 1.130 for miniTurbo-CCDC66. (p value of Welch's t test, \*\*p = 0.0043). Statistical analysis is performed on the means of 3 experimental replicates. The data underlying the graphs shown in the figure can be found in S1 Data. Noc, Nocodazole; TOGARAM, TOG array regulator of axonemal microtubule protein 1; MTs, microtubules; DMSO, dimethyl sulfoxide; MBP, Maltose-binding protein; BRB80, Brinkley Renaturing Buffer 80; DTT, Dithiothreitol; GTP, Guanosine triphosphate; RT, room temperature, CytoD, Cytochalasin D.

similar approach, we next investigated whether CEP104, a ciliary MAP previously reported to interact with CCDC66 during cilium assembly, could restore cilium disassembly and length phenotypes [55–58]. In IMCD3 cells stably expressing CEP104-BirA*-Flag (hereafter CEP104-BirA*), CCDC66 depletion led to faster cilium disassembly and shorter cilia compared to control cells (S2D–S2F Fig). These findings suggest two possibilities: either CCDC66 does not directly cooperate with CEP104 in regulating cilium length and disassembly in IMCD3 cells, unlike in RPE1 cells, or it acts upstream by recruiting CEP104 to microtubule ends [38,47].

The enhanced disassembly phenotype in CCDC66-depleted cells may result from destabilization of axonemal microtubules. To test this, we serum-stimulated ciliated control and CCDC66-depleted cells in the presence of 1 μM taxol to stabilize microtubules and then quantified the percentage of ciliated cells and cilium length 6 h after treatment (Fig 3C and 3D). Strikingly, taxol treatment reversed the deciliation phenotype in CCDC66-depleted cells but did not affect the reduction in cilium length, confirming that microtubule instability contributes specifically to increased cilium disassembly (Fig 3D–3F). Consistent with previous reports, taxol mildly promoted cilium disassembly in control cells, likely due to microtubule over-stabilization and decreased availability of soluble tubulin in the cytoplasm [24].

Finally, we investigated the relationship between CCDC66 and AURKA and HDAC6, two well-established regulators of cilium length, stability and disassembly [8,32,59]. To address this, we treated control and CCDC66-depleted cells with tubacin and MLN8237, specific inhibitors targeting HDAC6 deacetylase and AURKA kinase activities, respectively (Fig 3C). Each inhibitor effectively rescued the enhanced cilium disassembly observed in CCDC66-depleted cells at 6 h post-serum stimulation (Fig 3D and 3E). Although cilium length was reduced in both control and CCDC66-depleted cells at this time point, neither inhibitor restored the defects in cilium length (Fig 3D and 3F). These results show that inhibition of the AURKA/HDAC6 pathway activity specifically reverses disassembly defects associated with CCDC66 but does not correct length defects. Collectively, our findings show that CCDC66 inhibits cilium disassembly, potentially by stabilizing the axoneme, a process also regulated by the AURKA/HDAC6 pathway.

## CCDC66 depletion results in cilium length fluctuations and increased ectocytosis during disassembly

The processes underlying cilium disassembly within a single-cell population vary, including rapid deciliation via whole-cilium shedding (instant loss), gradual resorption, or a combination of both, where resorption precedes rapid deciliation [35]. To examine the role of CCDC66 in these distinct cilium disassembly events and the kinetics of disassembly, we monitored spatiotemporal dynamics of cilium disassembly in IMCD3::GFP-SSTR3 cells. Cells infected with control or Ccdc66-targeting virus were serum starved for 48 h, then serum-stimulated and imaged by confocal microscopy (Fig 4A). Quantification

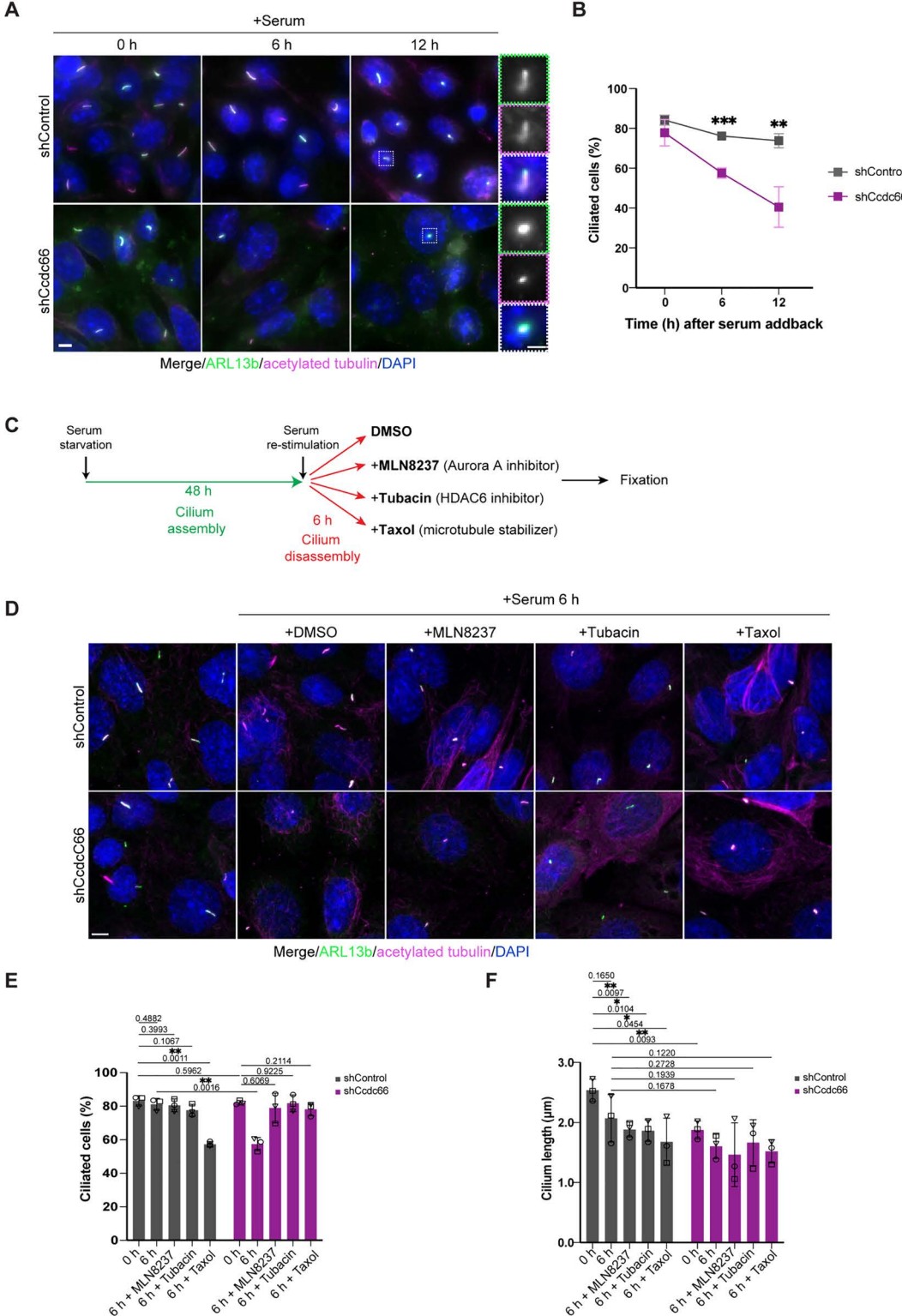

**Fig 3. Cilia disassembly proceeds at a faster rate in CCDC66-depleted cells, which is regulated by the AurkA-HDAC6 axis and rescued by microtubule stabilization. (A, B)** CCDC66 depletion leads to enhanced disassembly. Control and CCDC66-depleted cells were serum starved for 48 h, then serum stimulated for either 6 h or 12 h in total. Cells were fixed at indicated time points with 4% PFA, stained against anti-acetylated-tubulin,

ARL13b and DAPI, and imaged with widefield fluorescence microscopy. Scale bar: 5 µm. Insets show 3× magnifications of the cilia, Scale bar: 2 µm

**(B)** Quantification of ciliated cell percentage in **(A)**. Data represents mean±SD of 3 independent experiments. $n>600$ cells for control and >400 cells for CCDC66 depletion per experimental replicate. Mean cilia percentage at 0 h is 84.16% for shControl, 77.84% for shCcdc66; at 6 h is 76.15% for shControl, 57.56% for shCcdc66; at 12 h is 73.83% for shControl, 40.51% for shCcdc66. ($p$ values of multiple $t$ test of grouped data, ns not significant, ***$p=0.00032$, **$p=0.0059$). **(C, D)** AURKA and HDAC6 inhibition rescues enhanced cilium disassembly phenotype of CCDC66-depleted cells, as well as MT stabilization by Taxol **(C)** The experimental plan for rescue of cilia disassembly phenotype through inhibition of AURKA and HDAC6 and MT stabilization: control and Ccdc66 shRNA-depleted cells were serum starved for 48 h and released into serum-rich media to induce cilia disassembly for 6 h with DMSO (control), 500 nM AURKA inhibitor, MLN8237, 2 µM HDAC6 inhibitor, Tubacin, or 1 µM Taxol. Following incubation with inhibitors, cells were fixed and stained for acetylated-tubulin, ARL13b and DAPI. **(D)** Confocal microscopy of treated cells as in **(C)** with representation of CCDC66-depleted and treated cells compared to control cells, with quantification of cilia number and cilium length. **(E–F)** Data represents mean±SD of 3 independent experiments. $n>100$ cells for all control conditions, and on average 50–100 cells for all CCDC66-depletion conditions, per experimental replicate. The mean cilia percentage at 0 h is 83.22% for shControl, 82.07% for shCcdc66, at 6 h 81.12% for shControl, 57.36% for shCcdc66, 80.54% for shCCDC66 + MLN8237 and 78.96% for shCcdc66 + MLN8237, 77.59% for shControl + Tubacin and 81.75% for shCcdc66 + Tubacin, and 57.35% for shControl + Taxol and 78.16% for shCcdc66 + Taxol ($p$ values of one-way ANOVA with Tukey's post hoc test **$p=0.0012$). Cilia length was quantified in 3D according to ARL13b signal (co-localizing with acetylated-tubulin). The mean cilia length in CCDC66-depleted cells decreased to 0.74-fold compared to the mean control length at 0 h. Drug treatments decrease ciliary lengths with varying degree across replicates. $n>90$ cells for control treatments and >50 cells for CCDC66 depletion treatments per experimental replicate. $P$ values of multiple $t$ tests with Welch's corrections were represented together with the graph, $p$ values of one-way ANOVA with Welch's correction is *$p=0.0322$. Statistical analysis is performed on the means of 3 experimental replicates. The data underlying the graphs shown in the figure can be found in S1 Data. AURKA, Aurora kinase A; HDAC6, Histone deacetylase 6.

of cilium length during disassembly revealed a predominantly linear decrease in cilium length in control cells, whereas CCDC66-depleted cells displayed significant variations in length (Figs 4A, 4B, S3A and S3B). While cilia in control cells primarily disassembled through gradual resorption, CCDC66 depletion led to an increase in the frequency of instant loss, either independently or combined with gradual resorption (Figs 4A, 4C and S3A–S3C). Strikingly, about 40% of cilia in CCDC66-depleted cells disassembled within the first 2 h, classified as instant loss, compared to none in control cells (Figs 4B and S3A–S3C). Additionally, we observed faster cilium disassembly in CCDC66-depleted cells, with a mean disassembly rate of 1.03±0.53 µm/h ($n=30$), compared to 0.29±0.13 µm/h in control cells ($n=30$). Cilia exhibiting instant loss had the highest disassembly rates, leading to a wide range of cilium disassembly rates upon CCDC66 depletion (S3C Fig).

Next, we investigated the impact of CCDC66 depletion on ciliary ectocytosis during disassembly by quantifying ectocytosis events using live imaging videos. About 77.12% of the cilia in CCDC66-depleted cells underwent ectocytosis over 6 h of imaging, compared to less than 14.4% in control cells (Fig 4A and 4C). In a complementary analysis, we purified extracellular vesicles (EVs) from the supernatants of ciliated control and CCDC66-depleted IMCD3::GFP-SSTR3 cells induced for cilia disassembly with 6 h serum stimulation. Immunoblotting against GFP-SSTR3 and acetylated tubulin revealed a higher concentration of ciliary proteins in the EVs from CCDC66-depleted cells compared to control cells (Fig 4D). Collectively, these findings show that loss of CCDC66 enhances cilium disassembly and leads to cilium length fluctuations and increased ectocytosis, further supporting its functions in maintaining cilium instability.

### Temporal proximity mapping of CCDC66 during cilium disassembly reveals new mechanisms for its ciliary functions

To uncover new mechanisms underlying CCDC66 function during cilium disassembly, we identified its temporal proximity interactomes during this process. For these experiments, we used IMCD3 cells stably expressing miniTurbo or miniTurbo-CCDC66. Cells were serum-starved for 48 h, followed by induction of cilium disassembly and biotinylation (Fig 5A). They were then stained for Flag to mark the fusion protein, streptavidin to mark the biotinylated proteins, either ARL13b or acetylated tubulin to mark the cilium as well as PCM1 to mark the centriolar satellites (Fig 5A and 5B). Immunofluorescence analysis after 1 h and 6 h serum stimulation combined with 30 min biotinylation showed that miniTurbo-CCDC66 localized to and biotinylated proteins at centriolar satellites, basal body, and cilium. Notably, during cilium disassembly, the localization at centriolar satellites increased while localization at the cilium decreased (Fig 5A and 5B). In contrast, control cells expressing miniTurbo exhibited cytoplasmic biotinylation (Fig 5A).

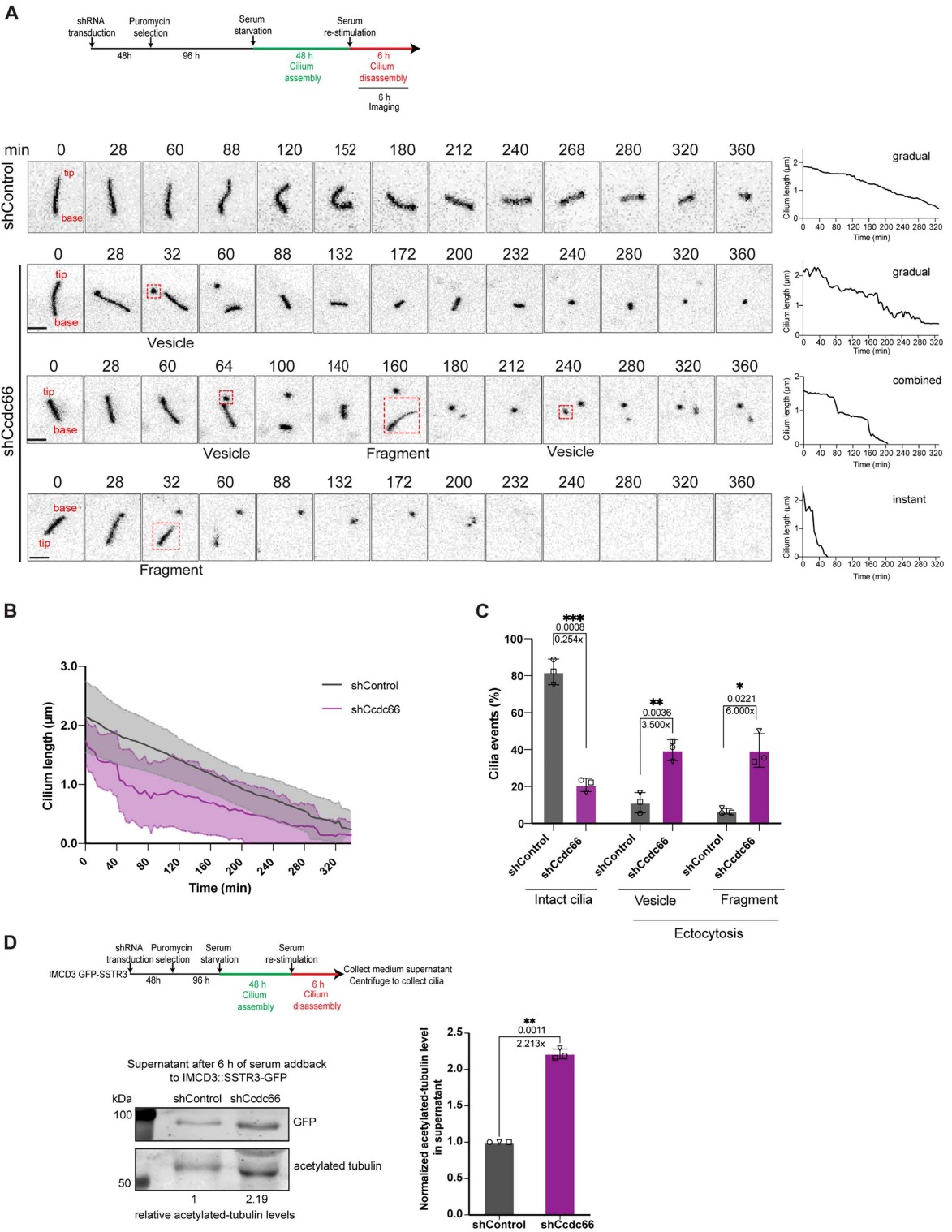

**Fig 4. Live imaging of CCDC66-depleted cells reveals increased extracellular vesicle and ciliary fragment shedding underlying fast cilia loss upon serum stimulation. (A–C)** Cilium disassembly kinetics upon serum stimulation. The IMCD3::SSTR3-GFP cells transduced with either control or Ccdc66 shRNA were grown and serum starved in FluoroDish for 48 h, then imaged with confocal microscopy for 6 h immediately upon serum

addition. Images were acquired every 4 min. Representative images show one control cilium that gradually resorbs, and three CCDC66-depleted cilia that undergo disassembly through gradual resorption, instant loss (whole cilium shedding), or a combination of both. Still images are inverted to better emphasize cilia. Scale bar 3 μm **(B)** Raw cilia length curves are measured from fluorescence of three independent experiments and represented as the mean ± SD. $n = 30$ cilia for both shControl and shCcdc66 conditions. **(C)** Quantification of ciliary ectocytosis events in **(A)**. Ciliary events, including vesicle releases (<500 nm) and cilium fragment shedding (>500 nm), were categorized based on the size of the released extracellular vesicles (EVs) and plotted as a bar plot ($p$ values of Welch's $t$ test \*\*\*$p = 0.0008$, \*\*$p = 0.0036$, \*$p = 0.0221$). **(D)** Media from IMCD3::SSTR3-GFP was collected 6 h post serum stimulation and separated by high-speed centrifugation for subsequent western blot. Pelleted material shows presence of ciliary markers at expected molecular weights suggesting pelleting of ciliary EVs and fragments. Membranes are blotted with anti-GFP and anti-acetylated tubulin. Shown in the graph is the fold change in the intensity of the acetylated tubulin bands, normalized to shControl of the displayed experimental replicate. The sample was normalized to total cellular protein abundance before loading to account for differences in cell numbers between control and CCDC66-depleted cells. $n = 3$. The bar plot represents the mean ± SD of 3 independent experiments. ($p$ values of Welch's $t$ test \*\*$p = 0.0011$). Statistical analysis is performed on the means of 3 experimental replicates. The data underlying the graphs shown in the figure can be found in S1 Data.

Following validation, we performed large scale streptavidin pulldowns from ciliated cells serum stimulated for 1 h and 6 h (S4A Fig), analyzed biotinylated proteins by mass spectrometry and defined high confidence CCDC66 interactome using Normalized Spectral Abundance Factor (NSAF) analysis [60] (S1 Table). This yielded 282 and 301 high-confidence interactors for CCDC66 after 1 h and 6 h serum time points, respectively, with 211 proteins common to both time points (S1 Table) (S4B Fig). We combined the proximity interactors from both time points to generate a comprehensive "CCDC66 disassembly interactome". We then analyzed this interactome using Gene Ontology (GO) analysis and literature mining and organized it into an interaction network in Cytoscape (Fig 5D). Among the enriched GO categories and functional clusters were microtubule organization, ciliogenesis, and cell division, consistent with the known functions of CCDC66 (Figs 5C, 5D, S4B and S4C). Notably, there was also enrichment of previously undescribed biological processes, including actin-binding proteins, and proteins involved in vesicle trafficking and lipid metabolism (Figs 5D and S4C).

We further cross-referenced the CCDC66 disassembly interactome with its ciliated proximity interactome, both generated using the same cell lines [38]. We reanalyzed the raw ciliated CCDC66 interactome using the same filtering thresholds that were applied to the disassembly interactome, to account for variations in methodologies and stringency levels. This revealed 202 proteins shared between ciliated and disassembly interactomes of CCDC66 (S4B Fig). For visualization of this comparative analysis, we plotted $\log_2$(NSAF) values in heat maps representing proteins linked to centrosome, cilia, centriolar satellites, actin, vesicles, and lipid metabolism (S4D–S4G Fig). Notably, we observed an enrichment of centriolar satellite proteins, which were reported for their functions during disassembly [61]. This enrichment corresponds with the redistribution of CCDC66 from the primary cilium to centriolar satellites during cilium disassembly (Fig 5B) [38]. Moreover, the interactome contained components of the CCDC66-linked Joubert syndrome module, such as ARMC9, TOGARAM1, CEP104 and CSPP1, known for their roles in cilium length regulation (Figs 5C, 5D and S4D) [45,51,56]. Similar to CCDC66, ARMC9/TOGARAM1 complex was shown to play a role in cilium stability and disassembly.

The identification of these validated interactors, linked to cilium stability, length regulation, and disassembly, supports the robustness of our interactome and highlights its potential for uncovering new mechanisms underlying CCDC66's role during cilium disassembly. Due to their abundance in the interactome and known associations with the cilium, we next investigated previously undescribed proximity relationships between CCDC66, the actin cytoskeleton, and vesicular trafficking.

## Actin cytoskeleton and vesicular trafficking proteins contribute to ciliary functions of CCDC66

Actin has been implicated in various ciliary processes, including ectocytosis, disassembly, length regulation and myosin VI-dependent protein transport to the cilium [24,25,62,63]. The overlap of these processes with ciliary roles of CCDC66, along with the enrichment of actin-binding proteins in the cilium disassembly interactome, suggests that CCDC66 may mediate its ciliary functions through interactions with actin, in addition to microtubules (Figs 5D and S4C) [62–64]. We showed that purified MBP-CCDC66 and mNG-MBP-CCDC66 bind to F-actin in vitro and resulting in reorganization of the

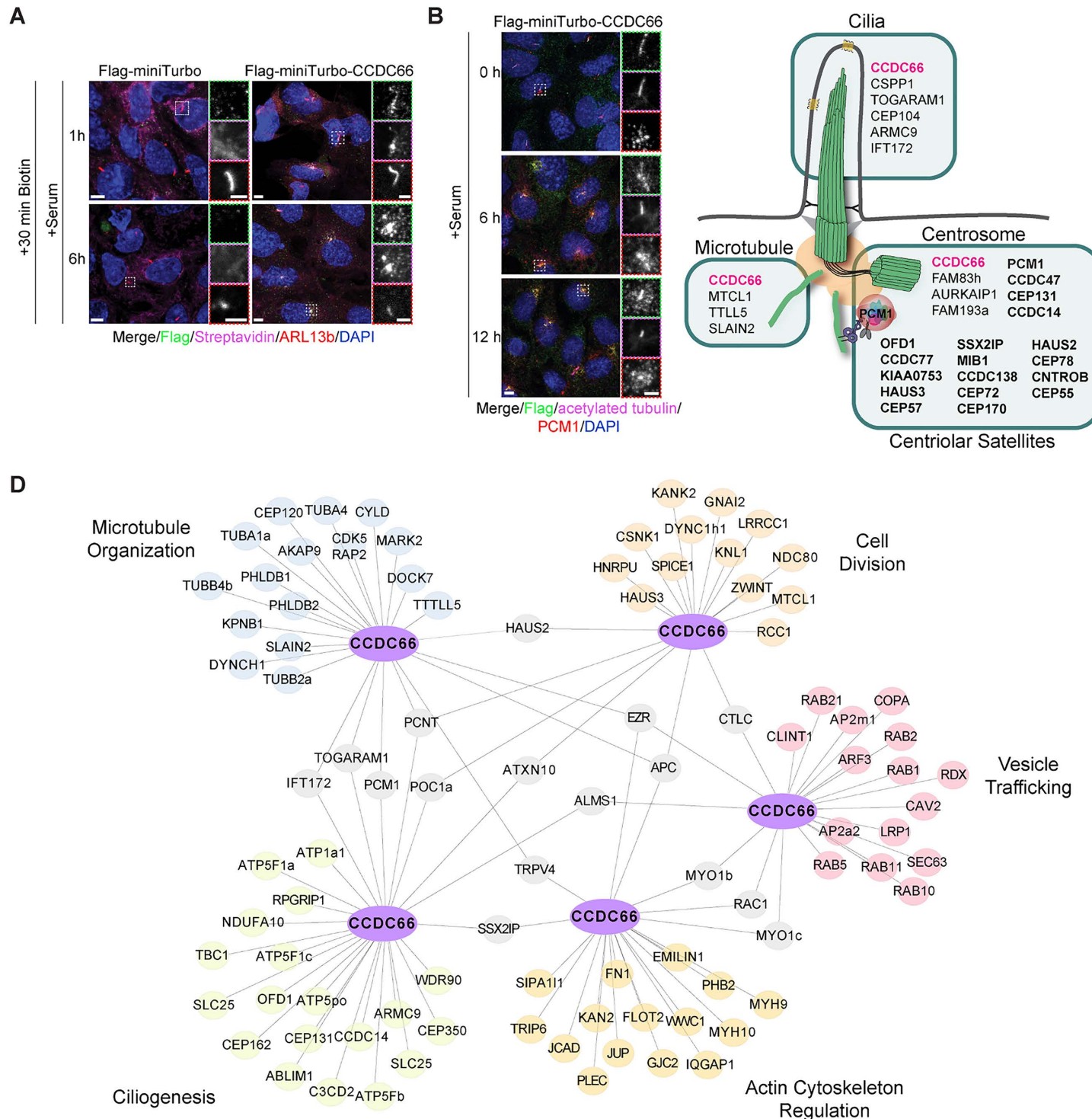

**Fig 5. Temporal proximity interactomes of CCDC66 during cilium disassembly. (A)** The ciliated Flag-miniTurboID or Flag-miniTurboID-CCDC66 IMCD3 cell lines were seeded on glass coverslips, treated with 500 µM biotin within the last 30 min of a 1 h or 6 h serum stimulation. Cells were then fixed with methanol and stained with anti-Flag, Streptavidin-Alexa568 coupled, anti-ARL13b, and DAPI, and imaged with confocal microscopy. Scale bar: 5 µm. Insets show 3× magnifications of the cilia, Scale bar: 2 µm. **(B)** CCDC66 translocates to centriolar satellites upon serum stimulation and cilia disassembly. IMCD3::Flag-miniTurboID-CCDC66 were fixed with 4% PFA at the indicated time points of serum stimulation and stained with anti-Flag, anti-PCM1, anti-acetylated-tubulin, and DAPI. Scale bar: 5 µm. Insets show 3× magnifications of the cilia, Scale bar: 2 µm. **(C)** Schematic diagram of

sub-interaction networks of CCDC66 interactome at 1 h and 6 h disassembly conditions based on cellular compartment. The CCDC66 proximity interactors were grouped using DAVID functional annotation tool and literature mining. The interaction networks visualized for the cellular compartments of the centrosome, cilia, and centriolar satellite were categorized to represent their respective unique compartments. Proteins highlighted in bold are the ones that localize to both centrosomes and centriolar satellites. **(D)** The sub-interaction networks of the CCDC66 interactome at 1 h and 6 h disassembly conditions. CCDC66 proximity interactors were grouped using the DAVID functional annotation tool and literature mining. The interaction networks visualized for cellular compartments include microtubule organization, cell division, ciliogenesis, vesicle trafficking and actin cytoskeleton regulation. The interconnectedness among the proteins in each network was determined using the STRING database. STRING, Search Tool for the Retrieval of Interacting Genes/Proteins; DAVID, Database for Annotation, Visualization, and Integrated Discovery; PCM1, pericentriolar material 1.

actin network (S5A and S5B Fig). Moreover, CCDC66 depletion led to changes in actin filament abundance and organization in IMCD3 cells (S5C Fig).

To further investigate this connection, we examined the role of actin in CCDC66 functions during cilium disassembly. First, we treated control and CCDC66-depleted IMCD3 cells with a low dose of CytoD and simultaneously induced cilium disassembly (Fig 6B). CytoD inhibited the enhanced cilium disassembly and shortening in CCDC66-depleted cells at 4 h post-serum induction (Fig 6A and 6C). To assess the impact of CytoD on cilium disassembly kinetics and ectocytosis events, we performed live imaging of control or CCDC66-depleted IMCD3::GFP-SSTR3 cells treated with CytoD (Fig 6D). CytoD treatment reduced cilium length fluctuations and slowed the disassembly rate in CCDC66-depleted cells (mean disassembly rate: $0.54 \pm 0.35$ μm/h, $n = 30$), compared to untreated CCDC66-depleted cells ($1.03 \pm 0.53$ μm/h, $n = 30$) and CytoD-treated control cells ($0.35 \pm 0.17$ μm/h, $n = 30$) (Figs 6E and S5D). These results suggest that actin-dependent processes, such as ectocytosis, may contribute to CCDC66 functions during cilium stability and disassembly.

Consistent with the inhibition of ectocytosis upon CytoD treatment, the percentage of intact cilia (those without vesicle ectocytosis or fragmentation during imaging) increased from about 20% in CCDC66-depleted cells to 50% in CCDC66-depleted cells upon CytoD treatment (S5E Fig). Concurrently, the percentage of ciliary ectocytosis decreased from about 40% to 23% upon CytoD treatment (Fig 6F). Despite this reduction, we still observed high levels of ciliary fragmentation, as assessed both by quantification of fragments from live imaging videos (about 27%) and from cells fixed and stained for ciliary markers (Fig 6F–6H). This suggests that some of these events might be driven by microtubule-induced destabilization of the axoneme.

Both actin and microtubule cytoskeleton modulate vesicular trafficking of cargo to the ciliary base, therefore regulating cilium homeostasis [65–67]. Specifically, various RAB proteins, including RAB8, RAB11, RAB5, and RAB7, have been shown to regulate axoneme extension, ciliary membrane formation, intraflagellar transport, cilium disassembly, and ciliary signaling [68]. Therefore, we investigated whether vesicular trafficking near cilia was disrupted by CCDC66 depletion during maintenance and disassembly. We first focused on the recycling and endocytic regulators RAB11 and RAB5, which were enriched in the disassembly proteome and are required for cilia formation and membrane composition (Figs 5D, S4C and S4F, and S1 Table) [54,67–69]. CCDC66 depletion significantly reduced the presence of RAB11 and RAB5-positive recycling and early endosomes near the ciliary base, particularly pronounced during cilium disassembly in the case of RAB11 (S6A–S6F Fig). These results suggest that CCDC66 may regulate early endocytic pathways, thereby promoting the directional trafficking of vesicles to the ciliary base.

The only RAB protein characterized for its role in cilium disassembly is RAB7, which has been shown to negatively regulate cilium ectocytosis and disassembly by controlling of intraciliary F-actin polymerization [49]. Since CCDC66 is involved in ciliary ectocytosis, we investigated its potential regulation of RAB7. In CCDC66-depleted cells, we observed an increase in RAB7-positive vesicles near the ciliary base, contrasting with reductions in RAB11 and RAB5 (S6G–S6I Fig). This suggests disruption of apical endocytosis and possible activation of protein degradation pathways. Total cellular levels of RAB11, RAB5 and RAB7 did not change upon CCDC66 depletion, confirming that observed changes are not due to their cellular availability (S6J Fig). Notably, we detected a few instances of RAB7 accumulation within CCDC66-depleted

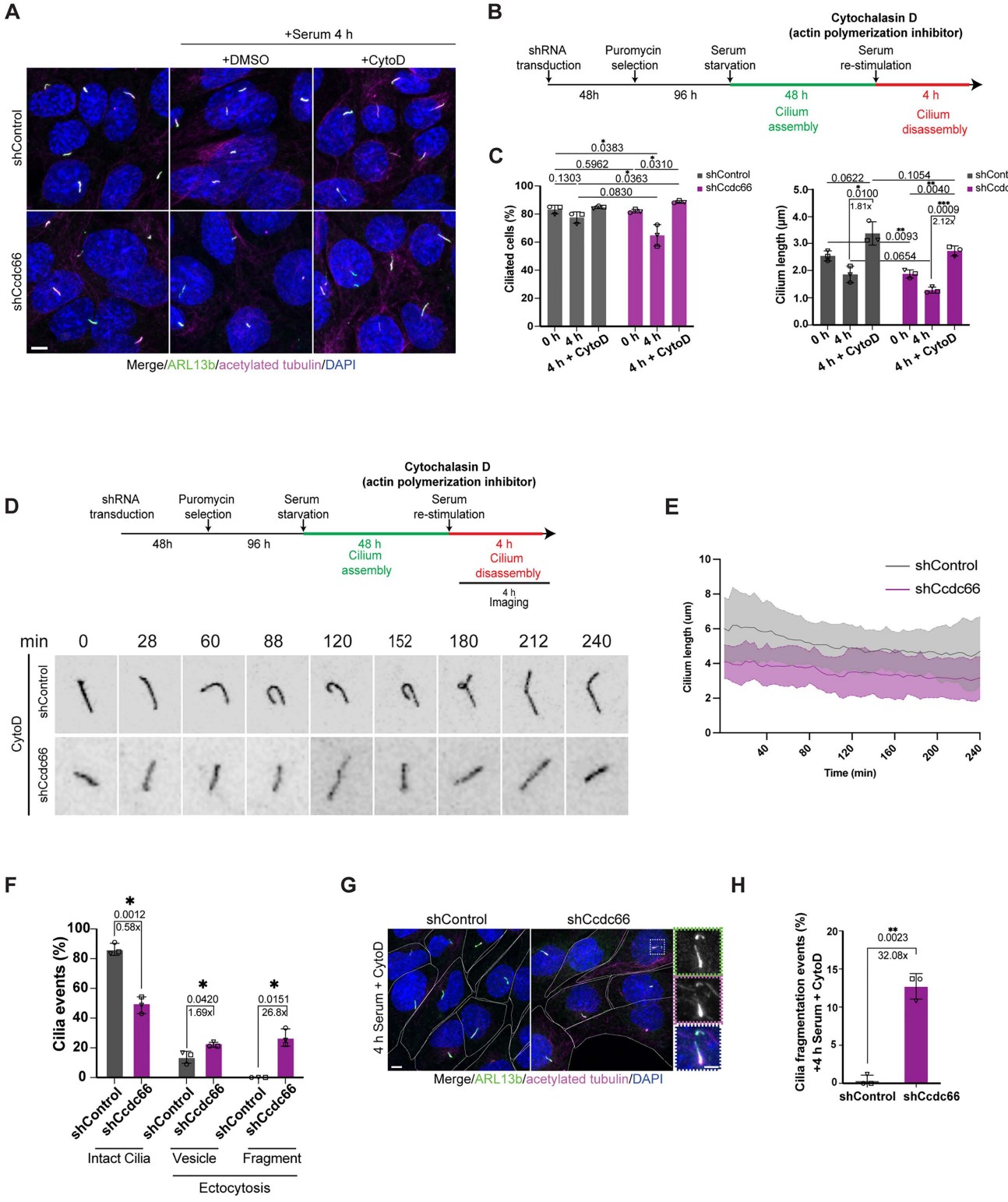

**Fig 6. Actin destabilization counteracts ectocytosis defects in CCDC66-depletion. (A)** Actin depolymerization with CytoD rescues enhanced cilium disassembly phenotype of CCDC66-depleted cells **(B)** The experimental plan for rescue of cilia disassembly phenotype through actin destabilization: control and Ccdc66 shRNA-depleted cells were serum starved for 48 h and released into serum-rich media to induce cilia disassembly for 4 h with DMSO (control), 0.5 μM Cytochalasin D. Following incubation with inhibitor, cells were fixed and stained for acetylated-tubulin, ARL13b

and DAPI. Confocal microscopy of treated cells with representation of CCDC66-depleted and treated cells compared to control untreated cells, with quantification of cilia number and cilium length. **(C)** Data represents mean±SD of 3 independent experiments. $n > 100$ cells for all control conditions, and on average 50–100 cells for all CCDC66-depletion conditions, per experimental replicate. The mean cilia percentage at 0 h is 83.22% for shControl, 82.07% for shCcdc66, at 4 h 77.51% for shControl, 64.81% for shCcdc66, 84.82% for shControl + CytoD and 88.65% for shCcdc66+CytoD ($p$ values are displayed above lines). **(C)** Cilia length was quantified in 3D according to ARL13b signal (co-localizing with acetylated-tubulin). The mean cilia length in CCDC66-depleted cells decreased to 0.74-fold compared to the mean control length at 0 h. Drug treatment increase ciliary lengths with similar degree in control and CCDC66-depleted cells, 1.8-fold increase in control and 2.1-fold increase in CCDC66 depletion. $n > 90$ cells for control treatments and >50 cells for CCDC66 depletion treatments per experimental replicate. **(D)** Cilium disassembly kinetics upon serum stimulation and simultaneous CytoD treatment. The IMCD3::SSTR3-GFP cells transduced with either control or Ccdc66 shRNA were grown and serum starved in FluoroDish for 48 h, then imaged with confocal microscopy for 4 h immediately upon serum addition together with CytoD. With images acquired every 4 min. Representative images show one control, and one CCDC66 depleted cilium maintaining length throughout imaging. Still images are inverted to better emphasize cilia. Scale bar 3 µm. **(E)** Raw cilia length curves are measured from fluorescence of three independent experiments and represented as the mean±SD. $n = 30$ cilia for both shControl and shCcdc66 conditions. **(F)** Quantification of ciliary ectocytosis events in **(D)**. Ciliary events, including vesicle releases (<500 nm) and cilium fragment shedding (>500 nm), were categorized based on the size of the released extracellular vesicles (EVs) and plotted as a bar plot ($p$ values of Welch's $t$ test **$p = 0.0012$, *$p = 0.042$, *$p = 0.0151$). **(G, H)** Cilia fragmentation event was also scored from fixed serum starved control and CCDC66-depleted cells, released into serum-rich media to induce cilia disassembly for 4 h with 0.5 µM Cytochalasin D. Cilia fragmentation events were scored per cell based on cell boundary drawn using cellular ARL13b and acetylated-tubulin signals, while cilia fragments were scored based on size >500 nm and positivity for ARL13b and proximity to cilia which it belonged to. The mean fragmentation percentage increased in CCDC66-depleted cells to 32-fold compared to the mean of control, $p$ value of Welch's $t$ test, **$p = 0.0023$. Data represents mean±SD of 3 independent experiments. $n > 50$ cilia for each cell line and treatment per experiment. The data underlying the graphs shown in the figure can be found in S1 Data.

cilia, leading to bending and decapitation (S6G Fig). This might explain the increased ectocytosis observed with CCDC66 depletion. Collectively, these results highlight potential new mechanisms through which CCDC66 may regulate actin dynamics and vesicular trafficking to maintain ciliary stability and integrity.

## CCDC66 depletion perturbs cilium content regulation and Hedgehog signaling

To investigate how ciliary defects linked to CCDC66 loss impact cilium function, we explored its role in Hedgehog signaling. Hedgehog ligand binding to PTCH1 triggers the phosphorylation and activation of the SMO receptor, facilitating its ciliary translocation and ultimately resulting in the transcriptional activation of Hedgehog target genes [70–72]. As readouts for Hedgehog pathway activation, we first quantified the ciliary entry efficiency of SMO and its active form, phospho-SMO (Fig 7A and 7B). To this end, ciliated control and CCDC66 depleted cells were treated with either with DMSO (control) or 1 µM Smoothened agonist (SAG) for 4 h and analyzed by immunofluorescence (Fig 7A). In both control and CCDC66-depleted cells, SMO and phospho-SMO did not localize to cilia under basal conditions (Fig 7A). After SAG stimulation, the percentage of SMO-positive cilia and the ciliary levels of SMO and phospho-SMO decreased significantly in CCDC66-depleted cells compared to control cells (Fig 7A and 7B). Moreover, we determined the downstream consequences of alterations in ciliary recruitment of SMO by quantifying Gli1 and Ptch1 transcriptional upregulation in basal conditions and SAG-treated cells. While SAG stimulation significantly upregulated Gli1 (about 3.4-fold) and Ptch1 (about 1.7-fold) in control cells, Gli1 showed a much smaller increase compared to control (about 1.5-fold), and Ptch1 showed no increase in CCDC66-depleted cells (Fig 7C). Notably, CCDC66 depletion also disrupted the basal level of regulation of these Hedgehog targets.

Defects in Hedgehog pathway activation following CCDC66 depletion might result from the destabilization of axonemal and cytoplasmic microtubules and subsequent alterations in ciliary processes. To investigate this, we examined whether stabilizing microtubules could reverse this phenotype. Notably, we observed increased recruitment of SMO, but not phospho-SMO, to the cilia in taxol-treated, CCDC66-depleted cells compared to untreated cells (S6K and S6L Fig). This demonstrates that CCDC66 and its microtubule-stabilizing function are important for Hedgehog pathway activation at the cilia.

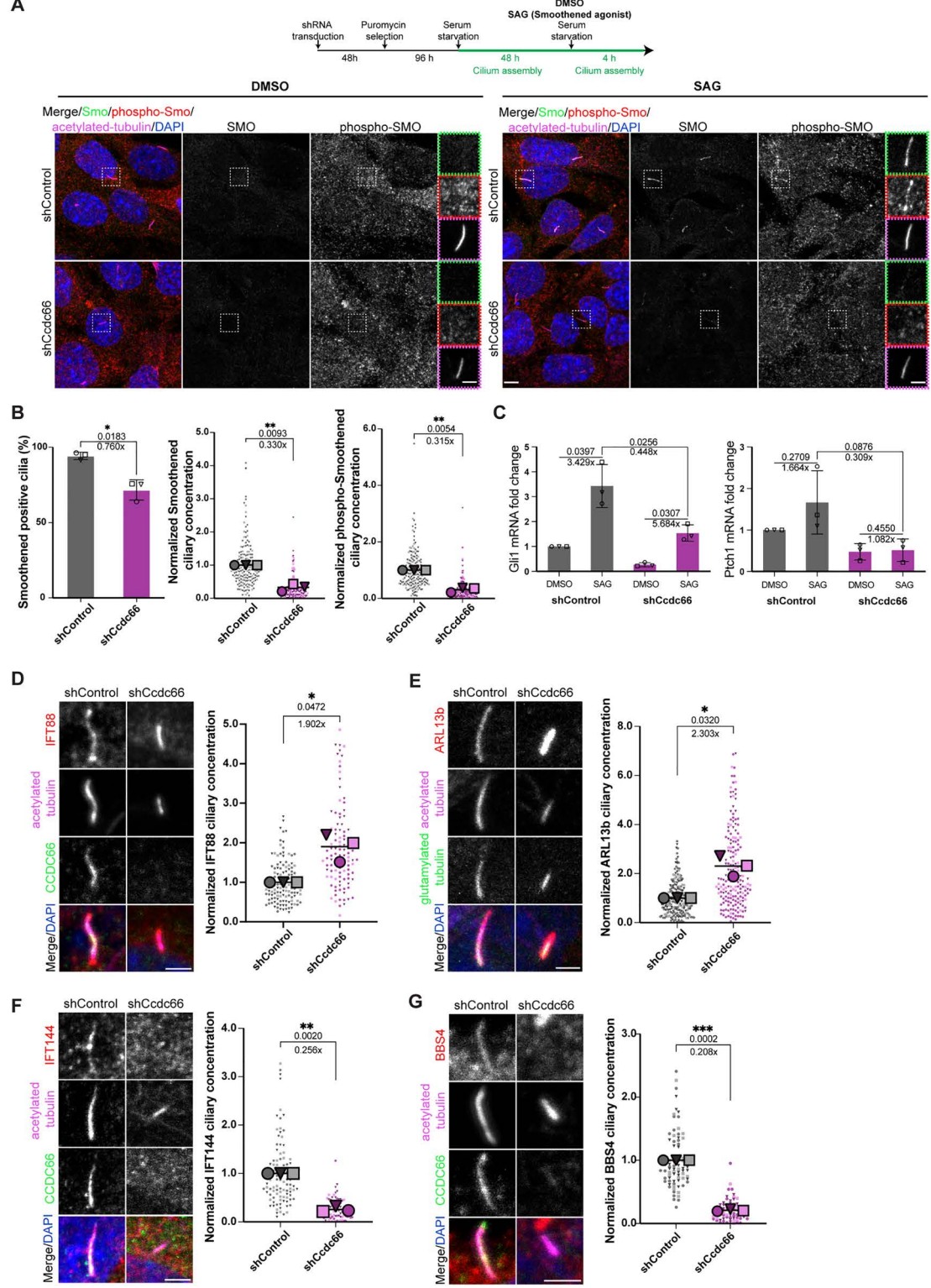

**Fig 7. CCDC66-depleted cells exhibit abnormal ciliary signaling and transport. (A, B)** Control and CCDC66-depleted cells were ciliated for 48 h, followed by treatment with 1 μM SAG or DMSO for 4 h, then fixed with 4% PFA and processed for immunofluorescence with anti-SMO, anti-phospho-SMO, anti-acetylated-tubulin and DAPI. Scale bar: 5 μm. Insets show 2.5× magnifications of the cilia, Scale bar: 2 μm **(B)** Quantification of SMO-positive cilia and ciliary SMO and phospho-SMO concentration in **(A)**. Data represents mean±SD for smoothened positive cilia and mean±SEM for ciliary

concentration of 3 independent experiments. $n > 100$ cells for control and CCDC66 depletion. Mean SMO-positive percentage is 94.19% for shControl and mean SMO-positive cilia is 67.72% for shCcdc66. ($p$ value of Welch's $t$ test, *$p = 0.0183$). Mean ciliary fluorescence density of SMO in CCDC66 depletion is decreased to 0.33-fold of the control mean. (Welch's $t$ test $p$ value **$p = 0.0093$). Mean ciliary fluorescence density of phospho-SMO in CCDC66 depletion is decreased to 0.315-fold of the control mean. (Welch's $t$ test $p$ value **$p = 0.0054$). **(C)** Effect of CCDC66 depletion on Hedgehog pathway activation as assessed by quantitative PCR. mRNA isolated from control and CCDC66-depleted cells, ciliated for 48 h and treated with either DMSO or SAG for 4 h, was analyzed with primers recognizing Gli1 and Ptch1, and β-actin primers as normalization control. Box plots show means ± SD. $n = 3$ technical replicates. Paired $t$ test $p$ values are presented above lines. **(D)** 48 h-ciliated control and CCDC66-depleted cells were fixed with 4% PFA and stained against anti-IFT88, anti-CCDC66 and anti-acetylated-tubulin. IFT88 signal was measured from CCDC66-depleted cells based on the absence or visible decrease of CCDC66 signal. Measured are ciliary integrated densities = mean intensity/area to represent ciliary signals of cilia varying in size. Super plot of normalized individual experimental values with mean± SEM represents 3 independent experiments. The different colored experimental replicates are shown as either circle, squares or triangles and with different color lightness. $n > 50$ cells for each condition. Mean ciliary fluorescence density of IFT88 in CCDC66 depletion is 1.90-fold higher than the control mean. (Welch's $t$ test *$p = 0.0472$). **(E)** Quantification of ciliary ARL13b in control and CCDC66-depleted cells, serum starved and stained against acetylated and polyglutamylated-tubulin and ARL13b to mark cilia. Measured are ciliary integrated densities. Super plot represents normalized individual experimental values and means ± SEM of 3 independent experiments. $n > 150$ cilia for each condition. (Welch's $t$ test *$p = 0.0320$). **(F)** Control and CCDC66-depleted cells were fixed with 4% PFA and stained against anti-IFT144, anti-CCDC66 and anti-acetylated-tubulin. Fluorescence intensity of IFT144 at the cilia were assessed as in **(D)**. Super plot of normalized individual experimental values with mean± SEM represents 3 independent experiments. $n = 100$ for control cells and $n = 73$ for CCDC66-depleted cells. Mean ciliary fluorescence density of IFT144 in CCDC66 depletion is decreased to 0.256-fold of the control mean. (Welch's $t$ test $p$ value ***$p = 0.0020$). **(G)** Control and CCDC66-depleted cells were fixed with 4% PFA and stained against anti-BBS4, anti-CCDC66 and anti-acetylated-tubulin. Fluorescence intensity of BBS4 at the cilia were assessed as in **(D)**. Super plot of normalized individual experimental values with mean ± SEM represents 3 independent experiments. $n > 50$ cells for each condition. Mean ciliary fluorescence density of BBS4 in CCDC66 depletion is decreased to 0.208-fold of the control mean. (Welch's $t$ test $p$ value ***$p = 0.0002$). Normalization in **(D–G)** is performed by dividing individual values with the average control value of experimental replicate. Statistical analysis is performed on the means of 3 experimental replicates. The data underlying the graphs shown in the figure can be found in S1 Data. SMO, Protein smoothened; SAG, Smoothened agonist; Gli1, Zinc finger protein GLI1; Pthc1, Protein patched homolog 1; IFT88, Intraflagellar transport protein 88 homolog; IFT144, Intraflagellar transport protein 144 homolog (WDR19, WD-repeat containing protein 19); BBS4, Bardet-Biedl syndrome 4 protein.

Additionally, defective ciliary signaling upon CCDC66 depletion may result from impaired intraciliary transport. Ciliary content, and consequently their signaling competence, is established by the intraflagellar transport (IFT) machinery involving IFT-A, IFT-B and BBSome complexes [1,73]. To investigate the impact of CCDC66 depletion on IFT, we assessed ciliary levels of IFT88 (anterograde) and IFT144 (retrograde). CCDC66-depleted cells showed an increased IFT88 and decreased IFT144, indicating IFT-A/IFT-B imbalance and disrupted ciliary transport (Fig 7D and 7F). Moreover, these cells exhibited increased ciliary levels of the small GTPase ARL13b, which is important for regulation of Hedgehog signaling and IFT (Fig 7E). Next, we measured the ciliary levels of BBS4, a component of the BBSome that redistributes between centriolar satellites and the primary cilium, similar to CCDC66. In CCDC66-depleted cells, the levels of BBS4 at the cilium were reduced compared to control cells, similar to previous observation in RPE1 cells [37] (Fig 7G). Together, these findings suggest that CCDC66 depletion disrupts ciliary signaling in part by compromising intraciliary transport.

## CCDC66 depletion perturbs epithelial cell organization

To investigate the consequences of cellular defects associated with CCDC66 loss in tissue architecture, we performed CCDC66 loss-of-function experiments using the 3D spheroid cultures of IMCD3 cells, which mimic in vivo organization of the kidney collecting duct [74–76]. To this end, control and CCDC66-depleted IMCD3 cells were cultured in Matrigel for 4 days, serum-starved for 2 days, and spheroid architecture was visualized by staining cells for markers for cilia (ARL13b), cell–cell contacts and polarity (actin and β-catenin). Control cells formed spheroids characterized by a central lumen, apically oriented cilia and well-organized apical and basal surfaces (Fig 8A). In contrast, CCDC66-depleted cells were impaired in their ability to form structurally proper spheroids compared to the control group (Fig 8A and 8B). We defined spheroids as defective if they exhibited any of the following characteristics: failure to form a lumen, formation of multiple or distorted lumens in 3D, nuclei protruding into the lumen, disorganized distribution of apical and basal markers, or misoriented cilia (Fig 8A and 8B). Analogous to the ciliogenesis defects of 2D serum-starved cultures, cilium length was reduced

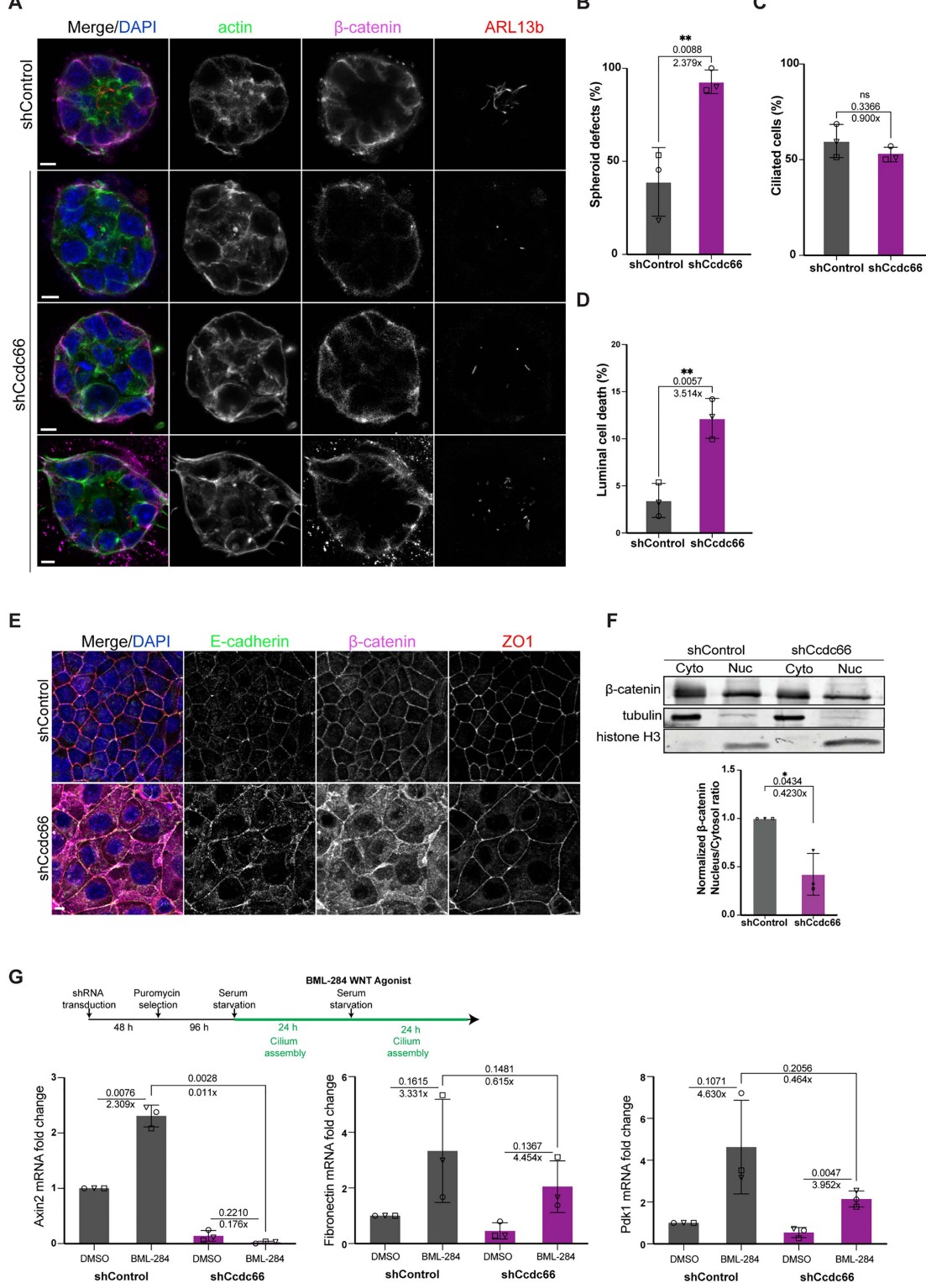

**Fig 8. The CCDC66 depletion disrupts normal tissue organization and signaling in 3D and 2D cultures. (A)** IMCD3 cells transduced and stably expressing either control shRNA or Ccdc66 shRNA were grown layered on 100% Matrigel for 4 days before transferring into serum-free media to induce ciliation for 48 h. Cells organized into epithelial spheroids in Matrigel, which were fixed with 4% PFA and processed for immunofluorescence analysis

with Phalloidin (staining F-actin), β-catenin and ARL13b, to observe apicobasal polarity and ciliation. Representative images show regular control spheroids and CCDC66-depleted spheroids with visibly disrupted apicobasal polarity, absence of lumen or cells protruding into lumen, as well as short misoriented cilia. Scale bar: 5 μm. **(B–D)** Quantification of spheroid defects in **(A)**. $n = 37$ spheroids for control and $n = 38$ spheroids for CCDC66 depletion. The mean defect percentage of three independent experiments is 38.99 for shControl and 92.72 for shCcdc66, with a fold change increase of 2.38. (*p* value of Welch's *t* test **$p = 0.0088$). **(C)** Quantification of cilia number in **(A)**. Data represents mean ± SD of 3 independent experiments. Mean cilia percentage is 59.78 for shControl and 53.80 for shCcdc66, with a fold change decrease of 0.90. (*p* value of Welch's *t* test ns: not significant). **(D)** Quantification of luminal cell death in **(A)**. The percentage of apoptosis was scored based on DAPI and β-catenin staining and distance of apoptotic bodies within spheroid lumen. Data represents mean ± SD of 3 independent experiments. Mean luminal cell death percentage is 3.46 for shControl and 12.16 for shCcdc66, with fold change increase in cell death of 3.51. (*p* value of Welch's *t* test **$p = 0.0057$). **(E)** Control and CCDC66-depleted cells were grown to 100% confluency to induce formation of cell-to-cell contacts, then fixed with 4% PFA and stained for tight-junction marker, ZO1, and adherens junction markers, E-cadherin and β-catenin. While tight junctions do not appear affected, adherens junction proteins display interrupted cell boundary localization, with increased cytoplasmic localization. Scale bar: 5 μm. **(F)** western blot analysis of cell lysates fractionated to separate cytosol and nuclear fraction from control and CCDC66-depleted samples then analyzed for difference in ratio of β-catenin in nuclear compared to cytosol fractions. Blots were stained with anti β-catenin antibody and anti-tubulin and anti-HistoneH3 as loading control for cytosolic and nuclear fractions, respectively. Represented is the ratio of fold changes of band intensities normalized to loading controls of displayed experimental replicates. $n = 3$. **(G)** Effect of CCDC66 depletion on Wnt pathway as assessed by quantitative PCR. mRNA isolated from control and CCDC66-depleted cells, ciliated for 24 h and threated either with DMSO or 700 nM Wnt agonist (BML-248) for 24 h, was analyzed with primers recognizing Axin2, PDK1 and Fibronectin, using β-actin primers as normalization control. Box plots show means ± SD. $n = 3$. Paired *t* test *p* values are presented above lines. Below *p* values presented are fold changes. The data underlying the graphs shown in the figure can be found in S1 Data. ZO1, Tight junction protein ZO-1; Cyto, cytosol fraction; Nuc, nuclear fraction; PDK1, Pyruvate dehydrogenase (acetyl-transferring) kinase isozyme 1; Wnt, Wingless-related integration site.

in CCDC66-depleted cells grown in 3D cultures while ciliation efficiency was not affected (Fig 8A and 8C). Additionally, as assessed by fragmented DNA, we found that CCDC66-depleted spheroids exhibited higher luminal death than control spheroids (Fig 8D).

Since cell-to-cell adhesion is a critical process required for proper spheroid formation, we next investigated cell-to-cell adhesion formation in 2D IMCD3 cultures. When grown to 100% confluency, IMCD3 cells typically form well defined cell contacts with apico-basal polarity (Fig 8E). However, CCDC66-depleted cells exhibited gross defects in cell-to-cell adhesion, as assessed by staining for E-cadherin and β-catenin, components of adherens junctions (Fig 8E). Cell-to-cell adhesion defects were less pronounced with the tight-junction marker ZO-1 (Fig 8E). Notably, cellular fractionation showed less accumulation of β-catenin in the nucleus compared to the cytoplasm in CCDC66-depleted cells relative to control cells (0.38-fold decrease) (Fig 8F).

Given the role of Wnt signaling in cell-to-cell adhesion along with our observation of disrupted β-catenin distribution upon CCDC66 depletion, we next investigated whether CCDC66 depletion affects this pathway [77,78]. To this end, we compared mRNA expression levels of Wnt target genes between control and CCDC66-depleted cells treated with control (DMSO) or Wnt agonist (BML-284). Specifically, we performed quantitative PCR analysis of one of the most responsive downstream target gene of β-catenin, and target genes involved in cell adhesion and cellular metabolism. As expected, BML-284 stimulation upregulated Axin2, Pdk1, and Fibronectin expression. However, CCDC66-depleted cells exhibited defects in the expression of Axin2, Pdk1, and Fibronectin compared to control cells in both basal and BML-284 stimulated conditions (Fig 8G). Collectively, these results indicate that CCDC66 plays an important role in maintaining cellular organization and adhesion in both 2D and 3D cultures, potentially due to its involvement in cilium maintenance, ciliary signaling and other non-ciliary functions.

## Discussion

CCDC66 is a MAP crucial for the proper formation and function of various microtubule-based structures and is linked to ciliopathies that affect the eyes and brain. In this study, we discovered new functions for CCDC66 in mouse kidney epithelial cells (Fig 9). First, it is required for maintaining the length, stability, and composition of steady-state and disassembling cilia. This involves regulating the stability of axonemal microtubules along with modulation of the actin cytoskeleton and the AURKA/HDAC6 cilium disassembly pathway. Second, CCDC66 regulate Hedgehog and Wnt signaling pathways,

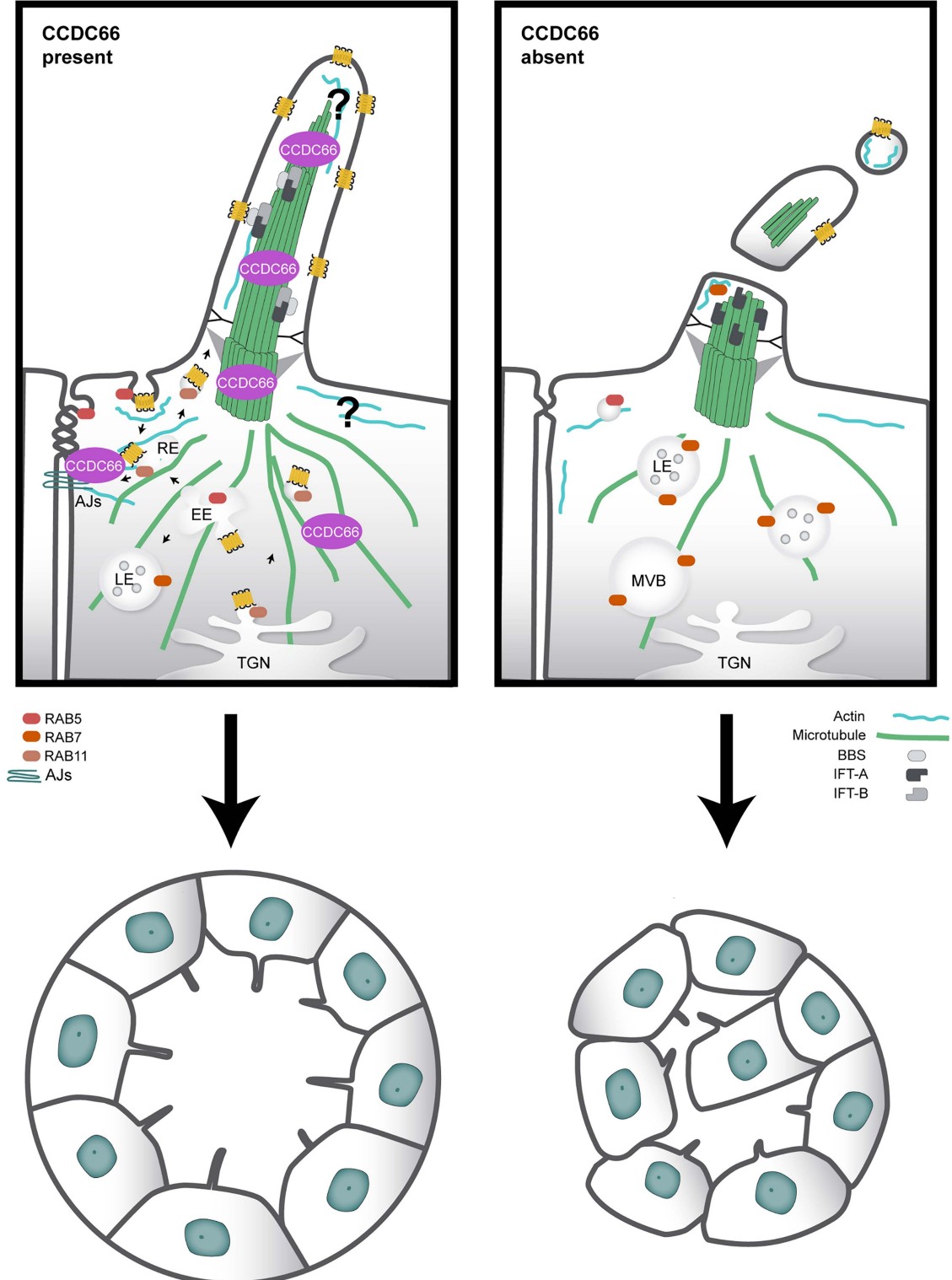

**Fig 9. Model for cellular functions and mechanisms of CCDC66.** CCDC66 is a microtubule-stabilizing protein that maintains cilium integrity and size and facilitates vesicular transport at the ciliary base. Depletion of CCDC66 results in variations in cilia size due to compromised axonemal integrity, intraflagellar transport, and increased ectocytosis events involving the release of vesicles and fragments. CCDC66 may also support ciliary ectocytosis

and length stability by regulating the actin cytoskeleton. Furthermore, disruptions in both microtubule stability and the actin cytoskeleton impair RAB11 and RAB5-mediated endocytosis and membrane recycling, affecting cell adhesion and the targeting of ciliary receptors. This disruption subsequently activates protein degradation pathways through RAB7-regulated late endocytosis, potentially involving RAB7 translocation to cilia to support ectocytosis. Collectively, these defects in protein transport, cilia function, and cell contacts contribute to compromised epithelial tissue integrity and multilumen formation in kidney tubules. Color Code Description: The color coding in the model is as follows: Cytoplasmic and axonemal microtubules are shown in green, actin in cyan, and intraflagellar transport-B (IFT-B) in dark blue. Light blue represents intraflagellar transport-A (IFT-A) and the BBSome complex membrane, while ciliary receptors such as Smoothened are highlighted in yellow. Adherens junctions (AJ) appear in navy blue. The gradient of greys indicates the components involved in recycling, early, and late endocytic trafficking. The recycling endosome (RE), marked by RAB11, and the early endosome (EE), marked by RAB5, are depicted in shades of pink, whereas late endosomes and multivesicular bodies (MVB), marked by RAB7, are illustrated in orange. Arrows indicate the typical, non-disrupted direction of vesicular trafficking from the *trans*-Golgi network (TGN) towards the cilia, plasma membrane, and cell contacts, as well as the membrane recycling route.

possibly through modulating intraciliary and vesicular transport. Importantly, these CCDC66 functions are necessary for tissue organization and cell polarity in 2D and 3D cultures. Our findings provide important insights into the maintenance and disassembly of the cilium and suggest that defects in cilium homeostasis, signaling, and tissue architecture contribute to the pathogenesis of ciliopathies associated with CCDC66.

Using loss-of-function and localization experiments, we identified CCDC66 as a critical regulator of different events of the cilium life cycle in IMCD3 cells. Although CCDC66 depletion impaired cilium elongation, it did not reduce overall ciliation efficiency in IMCD3 cells, unlike previously reported phenotypes in RPE1 cells. This discrepancy could reflect differential requirements for CCDC66 in intracellular versus extracellular ciliogenesis pathways, which assemble cilia with distinct structural properties [12,13,79]. Recent research has started to reveal these differences by identifying pathway-specific regulators, such as the Rab34 GTPase [80,81]. In addition to assembly, CCDC66 is required for maintaining cilium length and stability in steady-state cilia and during cilium disassembly. Defects in these processes upon CCDC66 depletion were reversed by inhibiting the AURKA-HDAC6 pathway, which destabilizes axonemal microtubules and taxol treatment, which stabilizes microtubules. Additionally, cilia expressing CCDC66 showed increased resistance to axoneme destabilization induced by microtubule depolymerization. Finally, in vitro microtubule stabilization experiments demonstrated that CCDC66 stabilizes microtubules and prevents their dilution-induced disassembly. These findings highlight microtubule stabilization as an important mechanism that underpins ciliary functions of CCDC66.

Cilium disassembly involves a complex interplay among ciliary transport, cytoskeletal dynamics, and intracellular signaling [8,82,83]. In agreement, the proximity interactome of CCDC66 during cilium disassembly identified diverse pathways and processes. To explore new mechanisms underlying CCDC66's functions, we focused on clusters containing actin-binding and vesicular trafficking-linked proteins. We observed that the frequent length fluctuations and enhanced cilium disassembly in CCDC66-depleted cells were partially restored by treatment with CytoD, which inhibited ciliary ectocytosis [23,53]. This suggests that CCDC66's ciliary functions are partially mediated by the processes involving actin, such as ciliary ectocytosis. We note that actin-mediated regulation of the cilium is multifaceted, involving different pools: the ciliary pool for ectocytosis, the cytoplasmic pool for myosin VI-dependent transport of ciliary proteins, and cortical actin for length regulation [24,25,62,63]. These findings imply that CCDC66 may regulate ectocytosis, actin-mediated trafficking, or cortical actin by modulating actin dynamics during cilium disassembly and length regulation. Additionally, given its direct affinity for both actin and microtubules, CCDC66 may regulate the ciliary cytoskeleton and related events by crosslinking microtubules with actin [36,46]. While dissecting the specific functions of the various cytoskeletal pools of CCDC66 will be challenging, future in vitro studies are expected to provide crucial insights into how CCDC66 regulates microtubule and actin dynamics.

We also showed that CCDC66's modulation of actin and microtubules may influence cilium homeostasis and signaling via regulation of vesicular trafficking at the ciliary base. CCDC66 depletion disrupted localization of specific RAB-marked vesicles near the base of cilia, which serves as a temporary reservoir for lipids and membrane proteins, facilitating their

delivery to the cilium [28,68,69,80,84,85]. While CCDC66 depletion led to a decrease in RAB5- and RAB11-marked early and recycling endosomes, it increased the presence of RAB7-positive lysosomes, MVB, and vacuoles. Although RAB7 was not detected in the CCDC66-proximity interactome, we propose that the observed increase in RAB7-positive vesicles is a physiological response to the reduction of RAB5-positive early endosomes following CCDC66 depletion. This is because the conversion of early endosomes into late endosomes depends on a shift from RAB5 to RAB7 [86,87]. The few instances of RAB7 accumulation within CCDC66-depleted cilia is also interesting as it might explain the increased ectocytosis phenotype. Future studies are needed to determine the extent to which the cytoskeletal and vesicular trafficking functions of CCDC66 contribute to its ciliary activities.

Characterization of CCDC66 during the activation of Hedgehog and Wnt pathways has revealed its roles in cellular signaling [2,77,88]. The functions and mechanisms of the primary cilium in Hedgehog pathway activation are well-established. Defects in this pathway are likely to result from impaired axonemal and cytoplasmic microtubule integrity due to the loss of CCDC66, which can subsequently impact vesicle trafficking, receptor targeting to cilia, and the intraciliary transport machinery. Regarding the Wnt pathway, our results provide the first evidence of CCDC66's involvement in Wnt signaling. We observed defects in the distribution of β-catenin in cells, particularly in its translocation to the nucleus. Consequently, there was a decrease in the transcriptional response of β-catenin's downstream targets. We note that the relationship between cilia and Wnt signaling remain less well-understood than Hedgehog signaling [77,78,89,90]. A significant body of research supports that primary cilia inhibit Wnt signaling. However, several signal-transducing proteins involved in the Wnt pathway, including Disheveled, FZD, Inversin, and GSK3, are also found within the cilium, suggesting ciliary WNT signaling [91–93]. While our results support this latter view, we also highlight that the defects observed in Wnt signaling might be a consequence of non-ciliary pools of CCDC66, such as those at cell-cell contact sites. The Wnt pathway has also been described for its roles in primary cilium assembly, with variations across different contexts and cell types [78,89]. Given the existing controversies, the extent to which we characterized the effect of CCDC66 on Wnt pathway activation is not sufficient to fully elucidate the nature of the relationship between CCDC66, Wnt signaling, and primary cilium assembly. Therefore, we have limited our analysis and conclusions to the functions of CCDC66 in Wnt signaling and its potential roles in cell-to-cell adhesion and epithelial cell organization.

Characterization of CCDC66 in both 2D and 3D IMCD3 culture models has revealed its roles in cell polarity and tissue organization. Proper cilium biogenesis and signaling are crucial for epithelial cell organization; therefore, it is expected that CCDC66 partly regulates these processes through its ciliary functions [74–76]. However, the contribution of non-ciliary functions cannot be excluded. Disrupted cell polarity and apical cell-cell adhesion, specifically at the adherens junctions (AJs), observed in both 2D and 3D cultures, suggest that CCDC66's role in organizing the cytoplasmic microtubule and actin cytoskeleton could underlie defective epithelial organization. A similar role in apical junction formation has been identified for ciliopathy proteins within the NPHP8-NPHP4-NPHP1 module [74,94]. Moreover, CCDC66's role in the canonical Wnt signaling pathway might also contribute to epithelial organization. This is further supported by studies showing that impaired Wnt signaling contribute to cell death and defective cell polarity [95–97]. Future studies using 3D cultures of human colon carcinoma Caco-2 cells or MCF10A mammary epithelial cells, which do not form cilia, will be necessary to differentiate between the ciliary and non-ciliary contributions of CCDC66 to epithelial cell organization.

In conclusion, CCDC66 emerges as a multifaceted regulator of ciliary and non-ciliary processes. In vitro and cellular studies have shown that CCDC66 regulates the abundance, organization and stability of microtubules in a context-dependent way [36–38,46]. These findings raise further questions about how CCDC66 regulates microtubules in its various roles, which could either converge on microtubule stabilization or diversify through the regulation of different microtubule properties and interacting with other cellular players. Supporting the latter, we discovered that CCDC66 plays a role in modulating the actin cytoskeleton and vesicular dynamics; however, the molecular mechanisms and direct consequences of these interactions remain to be explored. Finally, our study highlights the need to expand research on

CCDC66 mechanisms beyond retinal cells and documented retinopathies to include potential abnormalities in kidney development and other epithelial tissue defects. This would enhance our understanding of CCDC66-linked ciliopathies and could inform new therapeutic approaches for these and related diseases.

## Materials and methods

### Plasmids and primers

Mouse Ccdc66 shRNA (sequence 5′-ATCAGGGATCCACTTCTTAAT-3′), targeting nucleotides 2,635−2,655 of mouse Ccdc66, was cloned into pLKO.1 (Invitrogen) between the EcoRI and AgeI sites. The control shRNA (sequence 5′-CAA CAAGATGAAGAGCACCAA-3′) was previously described [87]. The plasmids pcDNA5.1-FRT/TO-Flag-miniTurboID-CCDC66 and pCDH-Flag-miniTurboID-Flag were previously described [42]. CEP104 cDNA (GenBank, NM_014704.4) was PCR amplified using forward primer 5′-GGGGACAACTTTGTACAAAAAAGTTGATatgccccacaagattggatttg-3′ and reverse primer 5′-GGGGACAACTTTGTACAAGAAAGTTGTgcgcttggcgtacgtcc-3′, cloned into pDONR221 and subsequently into pDEST-pcDNA5.1-FRT/TO-BirA*-Flag plasmid, which was provided by Anne-Claude Gingras (University of Toronto, Canada). Plasmids for protein expression in bacterial and insect cells, including pDEST-His-MBP, pDEST-His-MBP-mNeonGreen, pFastBac-HT-MBP-CCDC66, pFastBac-HT-MBP-mNeonGreen-CCDC66, and pFastBac-HT-MBP-msCCDC66 (clone 30626499), were previously described [40,42,88]. The pDONR221 plasmid containing the cDNA of human TOGARAM1 (GenBank, NM_001308120.2) was provided by Ronald Roepman (Radboud University Medical Center). The DNA fragment spanning the TOG1 and TOG2 domains of human TOGARAM1 (amino acids 2–621) was PCR amplified using primers forward 5′-GGGGACAACTTTGTACAAAAAAGTTGATctggcctcggccctc-3′, reverse 5′-GGGGACAACTTTGTACAAGAAAGTTGTtcaactctgagttctgttaccagcc-3′, and subsequently cloned into pDONR221 and then into pDEST-His-MBP vector for bacterial expression.

### Cell culture, transfection and transduction

Mouse Inner medullary collecting duct cell line (IMCD3) cells were cultured in Dulbecco's modified Eagle's Medium DMEM/F12 50/50 medium (Pan Biotech, Cat. # P04-41250) supplemented with 10% Fetal Bovine Serum (FBS, Life Technologies, Ref. # 10270-106, Lot # 42Q5283K) and 1% penicillin-streptomycin (GIBCO, Cat. # 1540-122). Human embryonic kidney (HEK293T, ATCC, CRL-3216) cells were cultured with DMEM medium (Pan Biotech, Cat. # P04-03590) supplemented with 10% FBS and 1% penicillin-streptomycin. All cell lines were authenticated by Multiplex Cell Line Authentication (MCA) and were tested for mycoplasma by MycoAlert Mycoplasma Detection Kit (Lonza). HEK293T cells were transfected with the plasmids using 1 mg/ml polyethylenimine, MW 25 kDa (PEI).

Recombinant lentiviruses were generated in HEK293T cells using pLKO.1 Scramble shRNA, pLKO.1 msCcdc66 shRNA, pCDH-miniTurboID-Flag plasmids as transfer vectors co-transfected with packaging and envelope vectors, psPax2 (Addgene #12260) and pMD2.G (Addgene #12259) using 1 mg/ml polyethylenimine, MW 25 kDa (PEI). Forty-eight hours after transfection, the supernatant was harvested, filtered, and titered on IMCD3 cells using GFP-expressing lentivirus. For transduction, IMCD3 cells were plated in 6-well tissue culture plates at $500 \times 10^5$, infected with an approximate 1 MOI over 2 days. For short-term depletion with shRNAs and stable long-term expression of fusion proteins, cells were selected with a medium containing 3 μg/mL puromycin medium for 4 days, then assayed 6–10 days after initial infection. The heterogeneous pools were used for functional assays. Depletion of the Ccdc66 mRNA and protein was confirmed by immunofluorescence, immunoblotting and qPCR.

### 3D spheroid assay

Forty μl/well of 100% Matrigel was solidified in Lab-Tek 8-well chamber slides (Thermo Fisher) by incubating for 15 min at 37°C. Control and CCDC66-depleted IMCD3 cells were trypsinized, washed with PBS and resuspended in DMEM/F12 50/50 supplemented with 2% fetal bovine serum and 2% Matrigel. Then, 5,000 cells/well were plated in Matrigel-coated

8-well chamber slides. Cells were cultured at 37°C for 4 days, serum-starved for 2 days, fixed with 4% PFA and processed for immunofluorescence by immunostaining with the indicated markers.

## Protein expression and purification

Protein expression and purification were performed as previously described [40]. BEVS baculovirus expression system and protocol was used for the expression of tagged full-length CCDC66 proteins [89]. Briefly, 100 mL of Hi5 cells ($1 \times 10^6$ cell/mL) were infected with P1 baculovirus produced in Sf9 cells, carrying His-MBP-mNeonGreen-CCDC66, His-MBP-CCDC66 or His-MBP-msCCDC66 at MOI of 1. Cells were collected 48 h after infection. For protein purification, bacterial or insect cells were lysed by sonication in lysis buffer (20 mM Hepes, pH 7.0 (or pH 7.5 for full-length protein), 250 mM NaCl, 0.1% Tween 20, 2 mg/ml lysozyme, 1 mM PMSF, 1 mM protease inhibitor cocktail, 5 mM BME and 10 mM Imidazole). His-tagged proteins were subsequently purified using Ni-NTA–agarose beads (Thermo Scientific, 88221) and elution buffer (20 mM HEPES, pH 7.0 or pH7.5, 250 mM NaCl, 5 mM BME, and 250 mM imidazole). For subsequent microtubule and actin assays, proteins were dialyzed against BRB80 buffer (80 mM PIPES, pH 6.8, 1 mM EGTA, 1 mM $MgCl_2$). For subsequent antibody production, proteins were dialyzed against PBS.

## In vitro microtubule stabilization assay

Stability assay by dilution was performed as previously described [90,91]. Briefly, MTs were polymerized at 2 mg/ml by incubating precleared tubulin in BRB80 with 1 mM DTT, 1 mM GTP and 25% glycerol for 30 min at 37°C. Following glycerol polymerization, MT were 10× diluted in BRB80 buffer with only DTT (no GTP and glycerol) and either with or without 100 nM of recombinant protein His-MBP-mNeonGreen, His-MBP-mNeonGreen-CCDC66 or His-MBP-TOGARAM1$_{TOG1TOG2}$. After 5 min incubation, reaction mixtures were fixed with 1% glutaraldehyde in BRB80 with DTT for 3 min, quenched with 100 mM Tris pH8.8 (final concentration) for 1 min, totaling to 11× dilution, then only half of the reaction was pelleted onto poly-L-lysine coated glass coverslips. MT control before dilution was immediately fixed with 1% glutaraldehyde in the volume of BRB80 with DTT, GTP, and glycerol to make the same 11× dilution, and half of the reaction was pelleted together with diluted samples. Centrifugation of fixed MT reactions was performed at 4,700 rpm for 45 min above 40% glycerol BRB80 cushion, fixed on coverslips with ice-cold methanol, and processed for immunofluorescence using anti-Tubulin antibody and confocal microscopy. The experiment was repeated 3 times.

## In vitro actin binding assay

Rabbit muscle actin was polymerized per manufacturer manual. Frozen 10 mg/ml stock was resuspended to 1 mg/ml with General Actin Buffer (20 mM Tris-HCl, pH 8.0, 0.2 mM $CaCl_2$) supplemented with 0.2 mM ATP and 1.0 mM DTT, and left on ice for 1 h. Polymerization of G-actin to F-actin was promoted by addition of 1/10th of the volume of 10× polymerization buffer (1 M KCl, 20 mM $MgCl_2$, 10 mM ATP) for 1 h at room temperature. Polymerized actin filaments were then diluted 10-fold in 1× polymerization buffer containing 70 nM Alexa 568 phalloidin, and bundling reactions were prepared. Labelled actin was incubated in F-Actin buffer (1× polymerization buffer supplemented with 1 mM ATP and 1 mM DTT) with 100 nM of recombinant protein, His-MBP, His-MBP-CCDC66, His-MBP-mNeonGreen or His-MBP-mNeonGreen-CCDC66 for 30 min at room temperature. The reaction mixtures were transferred into a flow chamber under a HCl-treated coverslip attached with double-sided tape to the microscope slide. Fluorescent proteins were observed with a Leica SP8 confocal microscope (Leica Mycrosystems). Experiment was repeated multiple times.

## Ultra-structure expansion microscopy (U-ExM)

U-ExM was performed as previously described [92] Briefly, IMCD3 cells were grown on glass coverslips and serum starved for 48 h. Coverslips were incubated in 1.4% formaldehyde/2% acrylamide (2× FA/AA) solution in 1× PBS for 5 h

at 37°C before gelation in Monomer Solution supplemented with TEMED and APS (final concentration of 0.5%) for 1 h at 37°C. Denaturation was performed at 95°C for 1 h 30 min and gels were stained with primary antibodies for 3 h at 37°C. Gels were washed three times for 10 min at RT with 1× PBS with 0.1% Triton-X (PBS-T) before secondary antibody incubation for 2 h 30 min at 37°C followed by three 10 min washes in PBS-T at RT. Gels were expanded by three exchanges of 150 ml dH$_2$O before imaging. The following reagents were used in U-ExM experiment: formaldehyde (FA), acrylamide (AA 40%), N,N′-methylenbisacrylamide (BIS, 2%), sodium acrylate (SA, 97%–99%), ammonium persulfate (APS), tetramethylethylendiamine (TEMED), and poly-D-Lysine.

## Immunofluorescence, antibodies and drug treatments

Cells were grown on coverslips, washed with PBS, and fixed by either 4% PFA diluted in cytoskeleton buffer (100 mM NaCl, 300 mM sucrose, 3 mM MgCl$_2$, and 10 mM PIPES) for 10 min at room temperature or ice-cold methanol at −20°C for 10 min. After washing three times with PBS, cells were blocked with 3% BSA in PBS plus 0.1% Triton X-100 for 30 min and incubated with primary antibodies in blocking solution for 1 h at room temperature. Cells were washed three times with PBS and incubated with secondary antibodies at 1:2000 dilution for 1 h and DAPI for 5 min at room temperature. Following three washes with PBS, cells were mounted using Mowiol mounting medium containing N-propyl gallate. Anti-CCDC66 antibody was generated by immunizing rats (Koc University, Animal Facility) with His-MBP-tagged mouse CCDC66 (clone 30626499) comprising amino acids 1–756 purified from Hi5 insect cells. The antibody was affinity purified against His–MBP–mCCDC66 (a.a. 1–756) and used at 0.5 µg/ml for immunofluorescence. Antibodies used for immunofluorescence in this study are detailed in S2 Table.

For microtubule depolymerization experiments, cells were treated with 400 ng/ml nocodazole or vehicle (dimethyl sulfoxide) for 2 h at 37°C. AURKA and HDAC6 inhibition was performed with 500 nM MLN8237 or 2 µM Tubacin for 6 h at 37°C. For microtubule stabilization experiments, cells were treated with 1 µM Taxol for 6 h at 37°C. For actin destabilization experiments and to prevent ciliary ectocytosis, cells were treated with 0.5 µM Cytochalasin D for 4 h (live cell experiments) at 37°C. Smoothened activation was induced by 1 µM SAG treatment for 4 h in serum-depleted IMCD3 cultures. Wnt pathway activation was induced by 700 nM Wnt Agonist BML-284 treatment for 24 h in serum-depleted IMCD3 cultures.

## Microscopy and image analysis

For the assessment of protein localization and level quantifications, images were acquired with Leica DMi8 fluorescent microscope with a stack size of 5 µm and step size of 0.25 µm in 1,024 × 1,024 format using an HC PL APO CS2 63 × 1.4 NA oil objective. Higher resolution images were taken by using an HC PL APO CS2 63 × 1.4 NA oil objective with Leica SP8 confocal microscope 1,024 × 1,024 pixel format, with pixel size ranging from 90 to 45 nm. For imaging spheroids, images were acquired with HC PL APO CS2 63 × 1.4 NA oil objective with Leica SP8 confocal microscope with a stack size of 50–70 µm and step size of 0.5 µm in 1,024 × 1,024 format with pixel size of 150 nm (above Nyquist sampling rate). For super-resolution imaging of endogenous CCDC66 localization, images were acquired using Elyra 7 with Lattice SIM$^2$ (Zeiss) and Zeiss Objective Plan-Apochromat 63×/1.4 Oil DIC M27, 633, 568 and 488 nm laser illumination, and standard excitation and emission filter sets. Sections were acquired at 0.110 µm z-steps. U-ExM samples were imaged using a Leica STELLARIS8 confocal microscope equipped with an HC PL APO CS2 63 × 1.4 NA oil objective, and the images were denoised using the Lightning Wizard. Time-lapse live imaging was performed with Leica SP8 confocal microscope equipped with an incubation chamber using HC PL APO CS2 63 × 1.4 NA oil objective. To image the effect of CCDC6 depletion on cilium maintenance, control or CCDC66-depleted IMCD3::GFP-SSTR3 cells were seeded at 1 × 10$^5$ cells/ml to FluoroDishes (WPI Europe). The following day, they were serum starved in DMEM/F12 without FBS for 48 h and subsequently imaged at 37°C with 5% CO$_2$. Imaging was performed with a frequency of 4 min per frame, with a 1 µm step size and a 12 µm stack size, in a 512 × 512 pixel format for 6 h. For observing cilium disassembly, control and

depleted cells were serum-starved for 48 h. After serum stimulation, cilium disassembly was imaged every 4 min for 6 h in a 512 × 512 pixel format. For observing the effect of CytoD on cilium disassembly, control and depleted cells were serum-starved for 48 h. After serum stimulation and simultaneous CytoD addition, cilium disassembly was imaged every 4 min for 6 h in a 512 × 512 pixel format. Images were processed using ImageJ (National Institutes of Health, Bethesda, MD).

The percentage of ciliated cells was determined by counting the total number of cells and the number of cells with primary cilia, as identified through DAPI staining of nuclei and either ARL13b or acetylated tubulin immunofluorescence of primary cilium. Ciliary length was measured in 3D using Aivia's automated object segmentation and measurement tools. Images were first preprocessed for background subtraction to enhance ciliary structures. Machine learning-based segmentation – 3D Object Analysis – Meshes module, was used to accurately identify and trace individual cilia based on ARL13b channel. Custom detection settings were applied: no smooth filtering, a minimum edge intensity threshold of 100, and custom partition settings featuring an object radius range of 0–50 μm and a mesh smoothing factor of 2. Following segmentation, the length of each cilium was quantified using Aivia's built-in measurement tools. The software's skeletonization and centerline extraction algorithms were employed to determine the precise length of each identified cilium. Manual verification was performed on a subset of images to ensure accuracy and consistency in automated measurements for the first two replicates. Statistical analysis of cilium length data was conducted using GraphPad Prism.

For quantitative immunofluorescence of ciliary and periciliary protein levels, z-stacks of cells were acquired using identical gain and exposure settings determined by the fluorescence signal in control cells. These z-stacks were used to create maximum-intensity projections using ImageJ (National Institutes of Health, Bethesda, MD). Ciliary regions were identified by the presence of ARL13b and/or acetylated tubulin signals. Periciliary regions were defined by ciliary ARL13b marker staining for each cell, and the mean pixel intensity of a circular 5 μm² area centered at the ciliary base in each cell was measured using ImageJ and defined as the periciliary intensity. Ciliary base was distinguished from the tip by ciliary thickness and proximity to nucleus. Background intensity was subtracted from the ciliary and periciliary fluorescence intensities. This subtraction was performed by quantifying fluorescence intensity in a region of equal dimensions in the area adjacent to the primary cilium or ciliary base. Ciliary protein concentration was determined by dividing the fluorescence signal of the protein to the cilium area, which was quantified using ARL13b or acetylated tubulin staining. Ciliary and periciliary protein concentrations and protein levels were normalized relative to the control group's mean for each experimental replicate (setting control mean to 1). For quantification of ciliary levels of proteins involved in ciliary transport (IFT88, IFT144, BBS4) and signaling components (SMO), we used anti-CCDC66 antibody, raised in rats, to identify CCDC66-depleted cells. We applied a quartile intensity threshold for the CCDC66 signal; measurements from cells above this threshold were excluded from our analysis.

Spheroids were manually analyzed to determine lumen formation and cell polarity based on staining for F-actin and β-catenin staining. Ciliary defects were assessed using ARL13b staining. Cell clusters smaller than 8 cells in cross-section or larger than 50 cells without a lumen were excluded from the analysis. With these criteria, approximately 50% of IMCD3 cells successfully formed spheroids. Among these, defective spheroids were defined as the ones that lacked a hollow lumen, had multiple or distorted lumens, showed disorganized apical and basal markers, and/or exhibited misaligned nuclei protruding into the lumen using the 3D rendering tool of LasX software (Leica Microsystems). Due to a large step size of 0.5 μm, which captured the entire size of the spheroids but omitted some cilia length details, cilia length measurements were not possible. Cilia count was based on ARL13b staining within regions stained for F-actin and β-catenin and was normalized by total cell count determined by DAPI staining to calculate the frequency of luminal cilia. Cell death was identified by counting apoptotic bodies (fragmented, condensed DNA based on DAPI intensity) and was limited to those within the spheroid lumen. This count was divided by the total cell number to calculate the frequency of luminal cell death.

For the analysis of live imaging movies, ciliary length was measured using a freehand line tool on 3D-rendered images in ImageJ. The quantification of ectocytosis events was performed manually by analyzing each cilium. The tips and bases of the cilia were identified using the cell's cytoplasm, marked by GFP-SSTR3, as a reference for the base. Fragments and vesicles were lost during imaging, while the cilia remained associated with the cell through its base. Fragment ectocytosis

was differentiated from vesicle ectocytosis by the size of the released ciliary piece; vesicle ectocytosis involved vesicles smaller than 0.5 μm, while fragment ectocytosis involved fragments larger than 0.5 μm.

The deciliation rate was calculated by measuring the difference between the initial and final cilium length and dividing by the time elapsed (hours). Data came from IMCD3::GFP-SSTR3 cells with either shControl or shCcdc66, with a total of 30 cilia analyzed across three biological replicates, as described in the analysis of live cell imaging of movies.

## Cell lysis and immunoblotting

Cells were lysed in RIPA buffer and protease inhibitors for 30 min at 4°C followed by centrifugation at 10,000 rpm for 30 min. For cell fractionation experiments, cells were resuspended in cytosolic lysis buffer (10 mM HEPES, pH 7.9, 10 mM KCl, 0.1 mM EDTA, 0.4% NP-40, plus fresh protease inhibitor cocktail). Cells were incubated on ice for 15 min followed by centrifugation at 3,000 × $g$ for 3 min at 4°C. Supernatant was collected and additionally centrifuged at 3,000 × $g$ for 5 min at 4°C. Collected supernatant was cytosolic fraction. Pellet from first centrifugation was washed with cytosolic lysis buffer, resuspended in nuclear lysis buffer (20 mM HEPES, pH 7.9, 0.4 M NaCl, 1 mM EDTA, 10% glycerol, with fresh protease inhibitor cocktail), and sonicated (2 × (10 s on 10 s off), amplitude 40%). Sample was then centrifuged at 15,000 × $g$ for 5 min at 4°C. Collected supernatant was nuclear fraction. The protein concentration of the resulting supernatants was determined with the Bradford solution (Bio-Rad Laboratories, CA, USA). For immunoblotting, equal quantities of cell extracts were resolved on SDS-PAGE gels, transferred onto nitrocellulose membranes and blocked with TBS with 0.1% Triton-X100 with 5% milk for 1 h at room temperature. Blots were incubated with primary antibodies diluted in 5% BSA in TBS-TX overnight at 4°C, washed with TBS-TX three times for 10 min and blotted with secondary antibodies for 1 h at room temperature. After washing blots with TBS-TX three times for 5 min, they were visualized with the LI-COR Odyssey Infrared Imaging System and software at 169 mm (LI-COR Biosciences). Antibodies used for Western blotting experiments used in this study are listed in S2 Table.

## RNA isolation, cDNA synthesis and qPCR

Total RNA was collected from control and CCDC66-depleted cells serum starved for 48 h to induce cilia formation. For Hedgehog pathway analysis, cells were also treated with 1 μM SAG 4 h before collection. For Wnt pathway activation analysis, cells were serum starved for 24 h, then treated with 700 nM Wnt Agonist BML-284 for 24 h in serum starvation medium before collection. RNA was isolated using the NucleoSpin RNA kit (Macherey-Nagel, Cat#: 740955.50) according to the manufacturer's protocol. The quantity and purity of RNA were assessed by measuring the optical density at 260 and 280 nm. Single-strand cDNA synthesis was performed with 1 μg of total RNA using SCRIPT Reverse Transcriptase (Jena Bioscience Cat#: PCR-511S) or iScript cDNA synthesis kit (BioRad, # 1708891). qPCR analysis of Ccdc66 mRNA levels was performed using GoTaq qPCR Master Mix (Promega) with primers 5′-GCAGAAAGCTGCCACAGAGA-3′ and 5′-CTGGGCTCTTCTTGCTTCCA-3′. Mouse Gapdh or β-actin was used as normalization controls with primers 5′-AAGGTCATCCCAGAGCTGAA-3′ and 5′-CTGCTTCACCACCTTCTTGA-3′ or 5′-GTTCGCCTTCATTATGGACTGCC-3′ and 5′-ATAGCACCCTGTTCCCGCAAAG-3′, respectively. Components of the Hedgehog signaling pathway, Gli1 and Ptch1, were analyzed using primers 5′-GCATGGGAACAGAAGGACTTTC-3′ and 5′-CCTGGGACCCTGACATAAAGTT-3′ and primers 5′-TGAACTGGGCAGCTATGAAGTC-3′ and 5′-ATGCTCCTTTCCTCCTGAAACC-3′, respectively. Downstream effectors of Wnt signaling using primers 5′-TGCGTTCTCGGAATAGCTCC-3′ and 5′-AGAGCTTTGCTGTAAAAGAGAGGA-3′ for Axin2, 5′-GGGCCAGGTGGACTTCTATG-3′ and 5′-CCACCGAACAATAAGGAGTGC-3′ for Pdk1 and 5′-AACTGGTTACCCTTCCACACC-3′ and 5′-TTCCAGGAACTTGGAACTGTAAGG-3′ for Fibronectin.

## Biotin identification experiments and mass spectrometry data analysis

IMCD3 cells stably expressing Flag-miniTurbo or Flag-miniTurbo-CCDC66 previously described [42] were used. For mass spectrometry analysis, each cell type was grown in 5 × 15 cm plates in DMEM/F12 medium supplied with 10%

FBS and 1% penicillin-streptomycin. The ciliated cell populations were generated after growing cells to 100% confluency and serum starving them for 48 h in DMEM/F12 with 0% FBS, followed by incubation with 500 μM biotin for 30 min. Cilium disassembly was induced in cells that had been serum-starved for 48 h, followed by serum stimulation for either 1 or 6 h. For biotinylation, cells undergoing disassembly were treated with 500 μM biotin for the last 30 min of the serum stimulation period. After the biotin treatment, cells were washed twice with PBS and lysed using RIPA buffer (50 mM Tris-HCl, pH 8.0, 150 mM NaCl, 0.1% SDS, 0.5% sodium deoxycholate, 1% Triton X-100), which was freshly supplemented with a protease inhibitor cocktail and 1 mM PMSF. Cell lysates were sonicated and then centrifuged at 16,000 × g for 1 h at 4°C. The resulting supernatant was incubated with Streptavidin–agarose beads (Thermo Fisher Scientific, Cat#: 20353) for 16 h at 4°C. After incubation, beads were washed twice with the RIPA buffer, once with 1 M KCl, once with 0.1 M $Na_2CO_3$, once with 2 M urea in 10 mM Tris-HCl pH 8.0 and finally twice with the RIPA buffer. For mass spectrometry analysis, the beads were resuspended in 100 μl of 50 mM ammonium bicarbonate and analyzed at the KUPAM proteomics facility as previously described [87]. The data presented in Figs 5 and S4, and S1 Table were obtained from two biological and two technical replicates each for Flag-miniTurbo CCDC66 ciliation, Flag-miniTurbo ciliation, and Flag-miniTurbo 1 h disassembly. For the Flag-miniTurbo CCDC66 1 h and 6 h disassembly, data were derived from two biological replicates, one with two technical replicates and the other with three technical replicates.

For mass spectrometry analysis, Normalized Spectral Abundance Factor (NSAF) values were calculated for each protein by dividing each Peptide Spectrum Match (PSM) by the total PSM count in that dataset. The datasets were filtered using several steps. First, proteins present only in the control dataset and in just one of the technical replicate datasets were excluded. NSAF values in the CCDC66 datasets were divided by the corresponding NSAF values from the control dataset to calculate an enrichment score, referred to as fold change hereafter. The average fold change was calculated for proteins present in both biological replicates. Proteins with an average $\log_2$(Fold Change) score less than 1 were removed. Next, the remaining proteins were analyzed using CRAPome (https://reprint-apms.org), a contaminant repository for mass spectrometry data from affinity purification experiments, where a list of contamination percentages (%) were generated [93]. Proteins with a contamination percentage greater than 50% were considered contaminants and removed. This cutoff value was selected based on the presence of any known interaction partners of CCDC66 within that range. Finally, proteins were organized according to their $\log_2$(Fold Change) values and an interactome map was generated using Cytoscape [94]. The edges of the high confidence hit proteins in the proximity map were plotted with the STRING database, and the map was visualized by CytoScape [95]. The functional clusters and GO categories for these clusters were determined with the Database for Annotation, Visualization and Integrated Discovery (DAVID) with a significance threshold of $P < 0.05$ and supplemented by literature mining [96].

Chemicals used throughout this study are listed in S2 Table

## Quantification and statistical analysis

Data were analyzed and plotted using GraphPad Prism 9 (GraphPad, La Jolla, CA). Results were presented as mean ± standard error of the mean (SEM) or mean ± standard deviation (SD). All experiments were performed in triplicate, and at least 50–300 cells per condition were analyzed to ensure statistical robustness. The number of biological replicates was indicated in the figure legends. Data were tested for normality, and appropriate statistical tests *t* test were applied to test a null hypothesis. Two-tailed unpaired *t*-tests, paired and unpaired *t*-tests with Welch correction (Welch's *t* test), one-sample *t*-tests, and one-way analysis of variance (ANOVA) were used to assess the statistical significance of measurements with a normal distribution. Data that did not follow a normal distribution were analyzed using a nonparametric Wilcoxon test. Normality was assessed using the Shapiro–Wilk Test. GraphPad built-in statistical test Rout 1% was performed to identify outliers in the case of ciliary concentration measurements in IFT88, ARL13b, acetylated and glutamylated-tubulin case. Statistical analyses were always performed on means of experimental replicates. The following

key is used for asterisks indicating *p*-values in the figures: \*p < 0.05, \*\*p < 0.01, \*\*\*p < 0.001, ns indicates not significant. Data underlying the graphs shown in all the figures can be found in S1 Data.

## Supporting information

**S1 Fig. CCDC66 affects cilium stability independently of the axonemal PTMs. (A–C)** The CCDC66 protein was successfully depleted from IMCD3 cells. Control and CCDC66-depleted cells were serum starved for 48 h, fixed with 4% PFA and stained against CCDC66 with homemade antibody, anti-acetylated-tubulin, and DAPI. Scale bars: 5 μm. Insets show 3× magnifications of the cilia and midbody (MB) signals. Scale bar 2 μm **(B)** western blot analysis of cell lysates from control and CCDC66-depleted samples using homemade CCDC66 antibody and mouse anti-GAPDH as loading control. Represented is fold change of band intensities normalized to loading control of displayed experimental replicates. *n* = 5 **(C)** qPCR analysis of mRNA isolated from control and CCDC66-depleted cells with primers recognizing C-terminal internal region and Gapdh primers as normalization control. Box plot shows mean ± SD. *n* = 4 (One sample *t* test *p* value \*\*p = 0.0006) **(D–F). (D)** Normalized data of cilia length quantification in Fig 1D. Normalization is performed by dividing individual length values with the average control value of experimental replicate **(E)** Normalized data of cilia length quantification in Fig 1E. Normalization is performed by dividing individual values with the average control value of experimental replicate at *t* = 0. **(F)** Quantification of ciliary axoneme PTMs in control and CCDC66-depleted cells, serum starved and stained against acetylated and polyglutamylated-tubulin and ARL13b to mark cilia. Measured are ciliary integrated densities = mean intensity/area, to better represent levels of PTMs in cilia of varying size. Super plot represents normalized individual experimental values and means ± SD of 3 independent experiments. *n* > 150 cilia for each conditioned and PTM. (*p* value of Welch's *t* tests ns: not significant). Statistical analysis is performed on the means of 3 experimental replicates. The data underlying the graphs shown in the figure can be found in S1 Data. MB, midbody; mRNA, messenger ribonucleic acid; GAPDH, Glyceraldehyde 3-phosphate dehydrogenase; PTM, post translational modification.
(TIF)

**S2 Fig. Cilium disassembly defects were rescued with ectopic expression of CCDC66 but not CEP104. (A–C)** IMCD3::Flag-miniTurboID- CCDC66 were transduced with control and CCDC66-targeting shRNA, selected and seeded on coverslips. After 48 h serum starvation, the cells were stimulated with serum-rich medium to induce cilia disassembly for total of 12 h, fixed and stained with anti-Flag, anti-acetylated-tubulin, anti-ARL13b and DAPI. Scale bar: 5 μm. Insets show 3× magnifications of the cilia, Scale bar: 2 μm **(B)** Quantification of cilia number in **(A)**. Data represents mean ± SD of 3 independent experiments. *n* > 400 cells for control and CCDC66 depletion at all three indicated time points. Mean cilia percentage at 0 h is 82.68% for shControl, 79.87% for shCcdc66; at 6 h is 80.52% for shControl, 80.0% for shCcdc66; at 12 h is 70.75% for shControl, 77.22% for shCcdc66. (*p* values of multiple *t* tests of grouped data *p* = 0.2457, *p* = 0.8590, *p* = 0.0651, ns: not significant). **(C)** Quantification of cilia length in **(A)** at 0 h. Super plot represents individual experimental values with mean ± SEM of 3 independent experiments. *n* > 400 cells for each condition. Mean cilia length measured in 3D in CCDC66 depletion is decreased to 0.97-fold of the mean control length. (Welch *t* test *p* value ns: not significant). **(D)** IMCD3::CEP104-BirA\*-Flag were transduced with control and CCDC66-targeting shRNA, selected and seeded on coverslips. After 48 h serum starvation, followed by serum addback for total of 12 h, fixed and stained with anti-Flag, anti-acetylated-tubulin, anti-ARL13b and DAPI. Scale bar: 5 μm. Insets show 3× magnifications of the cilia, Scale bar: 2 μm **(E)** Quantification of cilia number in **(D)**. Data represents mean ± SD of 3 independent experiments. *n* > 400 cells for control and CCDC66 depletion at all three indicated time points. Mean cilia percentage at 0 h is 85.22% for shControl, 86.75% for shCcdc66; at 6 h is 77.13% for shControl, 64.41% for shCcdc66; at 12 h is 74.54% for shControl, 54.53% for shCcdc66. (*p* values of multiple *t* test of grouped da*ta p* = 0.5229, \*\*p = 0.0084, \*p = 0.0141) **(F)** Quantification of cilia length in **(D)** at 0 h. Super plot represents individual experimental values with mean ± SEM of 3 independent. *n* > 400 cells for each

condition. Mean cilia length in CCDC66 depletion is decreased to 0.69-fold of the mean control length. (Welch's *t* test *p* value **$p = 0.0354$). Statistical analysis is performed on the means of 3 experimental replicates. The data underlying the graphs shown in the figure can be found in S1 Data. CEP104, Centrosomal protein of 104 kDa.
(TIF)

**S3 Fig. Live-cell imaging demonstrates that cilia disassembly predominantly occurs through combined and instant cilia loss in CCDC66-depleted cells. (A)** Individual raw cilia length connecting line curves from Fig 3B, measured from fluorescence of three independent experiments. $n = 30$ cilia for both shControl and shCcdc66 conditions. **(B)** Normalized data of cilia length quantification in Fig 3B. Normalization is performed by dividing individual values with the average control value of experimental replicate at $t = 0$. **(C)** Deciliation rate of cilia from Fig 3B (the speed of cilium shortening calculated as the length lost over time), measured from fluorescence of three independent experiments and represented as the mean $\pm$ SD. $n = 30$ cilia total for both shControl and shCcdc66 conditions. The data underlying the graphs shown in the figure can be found in S1 Data.
(TIF)

**S4 Fig. Gene ontology and comparative analysis of CCDC66 ciliated and disassembly proximity interactomes. (A)** Diagram depicting experimental plan of cell line treatments. Western blot analysis of samples from different stages of streptavidin pulldowns from IMCD3 cells expressing miniTurboID-Flag or miniTurboID-Flag-CCDC66. Cells were ciliated by serum starvation and then stimulated with serum for 1 h and 6 h. Samples were collected at the lysate (initial), pellet, and supernatant stages, and the non-eluted bead samples were run as the pulldown samples. Samples were blotted using Streptavidin-IRDye800 coupled and anti-Flag. Blue boxes show bands corresponding to Flag fusion proteins. **(B)** Comparison of the CCDC66 proximity interactomes of ciliation, 1 h disassembly and 6 h disassembly conditions. **(C)** GO-enrichment analysis of the CCDC66 disassembly interactome based on their biological process and cellular compartment. The *x*-axis represents the log-transformed *p*-value (Fisher's exact test) of GO terms. **(D–G)** Heat map showing $Log_2$(Fold Change) of the CCDC66 proximity interactors in ciliated cells versus 1 h and 6 h after serum stimulation of cilium disassembly. The range of the $Log_2$ Fold Change (FC) values is from 0 to 8, represented by shades of purple. Categories were determined using DAVID functional annotation tool and literature mining. Centrosome/cilia/centriolar satellite proteins were plotted in **(D)** where proteins with both centrosome and centriolar satellite compartment association are shown in bold. Actin-binding proteins were plotted in **(E)** where proteins shown in bold are ciliary actin-related proteins. Proteins linked to vesicular trafficking and lipid metabolism were plotted in **(F)** and **(G)**, respectively. The numerical data from the graphs shown in the figure can be found in S1 Data. GO, Gene Ontology.
(TIF)

**S5 Fig. Regulation of actin by CCDC66 is important for its ciliary functions. (A–B)** CCDC66 binds and bundles actin filaments in vitro. MBP, MBP-CCDC66 or MBP-mNG and MBP-mNG-CCDC66 were mixed with Alexa 568 Phalloidin stabilized polymerized actin in polymerization buffer and the reaction mixtures were loaded into a flow chamber under a HCl-treated coverslip attached with double-sided tape to the microscope slide. Fluorescent proteins and actin were observed with a Leica SP8 confocal microscope. **(C)** Effects of CCDC66 depletion on actin cytoskeleton. IMCD3 cells transduced and stably expressing either control or shCcdc66 were grown on glass coverslips, fixed with 4% PFA and imaged with confocal microscopy. Cells were co-stained with Sir-Actin and DAPI. Scale bar 5 µm. **(D)** Normalized cilia length curve from Fig 6E for both shControl and shCcdc66 conditions. Normalization is performed by dividing individual values with the average control value of experimental replicate at $t = 0$. **(E)** Quantification of the percentage of intact cilia based on fluorescence from three independent experiments, represented as the mean $\pm$ SD. A total of $n = 30$ cilia from 3 independent experiments were analyzed for both shControl and shCcdc66 conditions, either in the absence or presence of CytoD. The data underlying the graphs shown in the figure can be found in S1 Data.
(TIF)

**S6 Fig. The CCDC66 depletion disrupts endocytosis and membrane recycling near cilia. (A)** 48 h-ciliated control and CCDC66-depleted cells were fixed with 4% PFA and stained against anti-RAB5, anti-ARL13b and DAPI. Scale bar 5 μm. Insets show 1.3× magnifications of the cilia, Scale bar: 2 μm. **(B)** Measured is the mean pixel intensity of a circular 5 $\mu m^2$ area centered at the ciliary base of each cell in **(A)**. Super plot of normalized individual experimental values with mean ± SEM represents 3 independent experiments. Normalization is performed by dividing individual values with the average control value of experimental replicate. The different colored experimental replicates are shown as either circle, squares or triangles and with different color lightness. Approximately 100 cells for each condition. Mean periciliary levels of RAB5 in CCDC66 depletion is decreased to 0.387-fold of the control mean. (Welch's $t$ test ****$p < 0.0001$). **(C)** 48 h-ciliated control and CCDC66-depleted cells were treated with 0.5 μM CytoD for 4 h then fixed with 4% PFA and stained against anti-RAB11, anti-ARL13b and DAPI. Scale bar 5 μm. Insets show 1.3× magnifications of the cilia, Scale bar: 2 μm. **(D)** Measured is the mean pixel intensity of a circular 5 $\mu m^2$ area centered at the ciliary base of each cell in **(C)**. Super plot of normalized individual experimental values with mean ± SEM represents 3 independent experiments. The different colored experimental replicates are shown as either circle, squares or triangles and with different color lightness. Approximately 100 cells for each condition. Mean periciliary levels of RAB11 in CCDC66 depletion is 0.46-fold of the control mean. (Welch's $t$ test **$p = 0.0025$). **(E)** 48 h-ciliated control and CCDC66-depleted cells were serum stimulated and simultaneously treated with 0.5 μM CytoD for 4 h then fixed with 4% PFA and stained against anti-RAB11, anti-ARL13b and DAPI. Scale bar 5 μm. Insets show 1.3× magnifications of the cilia, Scale bar: 2 μm. **(F)** Measured is the mean pixel intensity of a circular 5 $\mu m^2$ area centered at the ciliary base of each cell **(E)**. Super plot of normalized individual experimental values with mean ± SEM represents 3 independent experiments. Normalization is performed by dividing individual values with the average control value of experimental replicate. The different colored experimental replicates are shown as either circle, squares or triangles and with different color lightness. Approximately 100 cells for each condition. Mean periciliary levels of RAB11 in CCDC66 depletion is 0.46-fold decreased than the control mean. (Welch's $t$ test ***$p = 0.0004$). **(G)** 48 h-ciliated control and CCDC66-depleted cells were fixed or serum stimulated for 6 h then fixed with 4% PFA and stained against anti-RAB7, anti-ARL13b and DAPI. Scale bar 5 μm. Insets show 3.6× magnifications of the cilia, Scale bar: 2 μm. **(H)** Measured is mean pixel intensity of a circular 5 $\mu m^2$ area centered at the ciliary base of each cell after 48 h ciliation. Super plot of normalized individual experimental values with mean ± SEM represents 3 independent experiments. The different colored experimental replicates are shown as either circle, squares or triangles and with different color lightness. Approximately 100 cells for each condition. Mean periciliary levels of RAB7 in CCDC66 depletion at 0 h is 6.66-fold increased than the control means. (Welch's $t$ test **$p = 0.0021$) **(I)** Measured is the mean pixel intensity of a circular 5 $\mu m^2$ area centered at the ciliary base of each cell after 48 h ciliation and 6 h serum addback. Mean periciliary levels of RAB7 in CCDC66 depletion at 6 h is increased 9.19-fold of the control mean. (Welch's $t$ test *$p = 0.0334$). **(J)** western blot analysis of samples from IMCD3 cells transduced with shControl or shCcdc66. Cells were ciliated by serum starvation and the initial 0 h sample of ciliated cells were collected. The cells stimulated with serum for 1 h and 6 h were also collected. Samples were run on SDS-PAGE and analyzed by immunoblotting using anti-GAPDH for loading control and anti-CCDC66, anti-RAB11, anti-RAB7, anti-RAB5 antibodies. **(K)** Taxol stabilization of microtubules partially rescues SMO response in CCDC66 depleted cilia. Control and CCDC66-depleted cells were ciliated for 48 h, followed by treatment with 1 μM SAG for 4 h or with 1 μM SAG together with 1 μM Taxol, then fixed with 4% PFA and processed for immunofluorescence with anti-SMO, anti-phospho-SMO, anti-acetylated-tubulin and DAPI. Scale bar: 5 μm. Insets show 2.5× magnifications of the cilia, Scale bar: 2 μm. **(L)** Super plots represent normalized individual experimental values and means ± SEM of 3 independent experiments. Normalization is performed by dividing individual values with the average control value of experimental replicate. Mean ciliary fluorescence density of SMO in CCDC66 depletion after Taxol treatment is decreased to 0.72-fold of the control mean (compared to 0.33-fold without Taxol in Fig 7). (Welch's $t$ test $p$ value **$p = 0.0011$). Mean ciliary fluorescence density of phospho-SMO in CCDC66 depletion is decreased to 0.31-fold of the control mean. (Welch's

*t* test *p* value **$p = 0.0024$). Box plot represents mean $\pm$ SD for smoothened positive cilia in SAG-treated control and CCDC66-depleted cilia without and with Taxol treatment. Paired *t* test *p* values are presented above lines. The data underlying the graphs shown in the figure can be found in S1 Data. RAB5, Ras-related protein RAB5; RAB7, Ras-related protein RAB7; RAB11, Ras-related protein RAB11.
(TIF)

**S1 Table.  Mass spectrometry results related to Figs 5 and S4.** Column explanations were placed to sheet named "Legend".
(XLSX)

**S2 Table.  Includes catalog number for all chemicals and antibodies used in this study.**
(XLSX)

**S1 Data.**   Numerical data for Figs 1C, 1D, 1F, 1G, 2B, 2E, 3B, 3E, 3F, 4A–4D, 6C, 6E, 6F, 7B–7G, 8B–8D, 8F, 8G, S1C–S1F, S2B, S2C, S2E, S2F, S3C, S3A–S3C, S4C–S4F, S5D, S5E, S6B, S6D, S6F, S6H, S6I, S6L. Each tab in the file represents the corresponding figure panel produced based on the displayed data.
(XLSX)

**S1 Raw Images.  Unprocessed images for blots shown in the paper.** Red boxes are used to show the represented blot in the figures. Red crosses show the parts of the blot that are not used in the figures.
(TIF)

## Acknowledgments

We acknowledge Melis Dilara Arslanhan, Ezgi Odabasi and Efe Begar for their insightful feedback on this work. We thank Melis Dilara Arslanhan Gül for help with U-ExM. We also acknowledge use of the services and facilities of the KUPAM – Koç University Proteomics Facility. We thank Dr. Attila Gürsoy's lab for the script which helped us combine mass spec facility data.

## Author contributions

**Conceptualization:** Jovana Deretic, Elif Nur Firat-Karalar.

**Data curation:** Jovana Deretic, Seyma Cengiz-Emek, Ece Seyrek.

**Formal analysis:** Jovana Deretic, Seyma Cengiz-Emek, Ece Seyrek.

**Funding acquisition:** Jovana Deretic, Elif Nur Firat-Karalar.

**Investigation:** Jovana Deretic, Elif Nur Firat-Karalar.

**Methodology:** Jovana Deretic, Seyma Cengiz-Emek, Ece Seyrek, Elif Nur Firat-Karalar.

**Project administration:** Jovana Deretic, Elif Nur Firat-Karalar.

**Resources:** Jovana Deretic, Elif Nur Firat-Karalar.

**Software:** Seyma Cengiz-Emek, Elif Nur Firat-Karalar.

**Supervision:** Jovana Deretic, Elif Nur Firat-Karalar.

**Validation:** Jovana Deretic.

**Visualization:** Jovana Deretic, Ece Seyrek, Elif Nur Firat-Karalar.

**Writing – original draft:** Jovana Deretic, Elif Nur Firat-Karalar.

**Writing – review & editing:** Jovana Deretic, Seyma Cengiz-Emek, Ece Seyrek, Elif Nur Firat-Karalar.

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
