## [Editor Report · Decision Letter 0]

Dear Elif,

Thank you for submitting your manuscript entitled "Ccdc66 regulates primary cilia stability, disassembly and signaling important for epithelial organization" for consideration as a Research Article by PLOS Biology.

Your manuscript has now been evaluated by the PLOS Biology editorial staff as well as by an academic editor with relevant expertise and I am writing to let you know that we would like to send your submission out for external peer review.

Once your full submission is complete, your paper will undergo a series of checks in preparation for peer review. After your manuscript has passed the checks it will be sent out for review. To provide the metadata for your submission, please Login to Editorial Manager (https://www.editorialmanager.com/pbiology) within two working days, i.e. by Jun 10 2024 11:59PM.

Kind regards,

Ines

--

Ines Alvarez-Garcia, PhD

Senior Editor

PLOS Biology

---

## [Decision Letter · Decision Letter 1]

Dear Elif,

Thank you for your patience while your manuscript entitled "Ccdc66 regulates primary cilium stability, disassembly and signaling important for epithelial organization" was peer-reviewed at PLOS Biology. Please also accept my apologies for the delay in providing you with our decision. Your manuscript has been evaluated by the PLOS Biology editors, an Academic Editor with relevant expertise, and by three independent reviewers.

The reviews are attached below. As you will see, the reviews find the conclusions potentially interesting, however they also raise several concerns and suggest several experiments that are needed to confirm the findings before we can consider the manuscript for publication. They all find the experiments showing a potential function for Ccdc66 weak and not suitable to distinguish between a direct role on ciliary disassembly or on ciliary length regulation. In addition, Reviewer 1 suggests an alternative interpretation of the data, and Reviewer 2 thinks that the statistical analysis should improve. Reviewer 3 notes that the analysis of the functional consequences of Ccdc66 depletion in dampened Hh and Wnt response needs some work and suggests several experiments to strengthen these results.

Based on the reviewers' comments and following discussion with the Academic Editor, it is clear that a substantial amount of work would be required to meet the criteria for publication in PLOS Biology. However, given our and the reviewer interest in your study, we would be open to inviting a comprehensive revision of the study that thoroughly addresses all the reviewers' comments.

We appreciate that these requests represent a great deal of extra work, and we are willing to relax our standard revision time to allow you 6 months to revise your study. Given the extent of revision that would be needed, we cannot make a decision about publication until we have seen the revised manuscript and your response to the reviewers' comments. Your revised manuscript would need to be seen by the reviewers again, but please note that we would not engage them unless their main concerns have been addressed. Please email us (plosbiology@plos.org) if you have any questions or concerns, or envision needing a (short) extension.

**IMPORTANT - SUBMITTING YOUR REVISION**

3. Resubmission Checklist

a) *PLOS Data Policy*

b) *Published Peer Review*

Sincerely,

Ines

--

Ines Alvarez-Garcia, PhD

Senior Editor

PLOS Biology

Reviewers' comments

Rev. 1:

Deretic et al claimed that Ccdc66 is a microtubule stabilizing factor in epithelial tissues and is essential for primary cilium maintenance and disassembly. In this manuscript, the authors observed the effect of Ccdc66 depletion on each process of cilia assembly, maintenance, and disassembly in IMCD3 cell line using several molecular/genetic tools. Additionally, the authors sought to demonstrate the essential role of Ccdc66-mediated cilia in epithelial organization through the control of Shh/Wnt signaling.

Although these observations, and particularly the cilia disassembly-specific data, are interesting, the logic and evidence for claiming a role for Ccdc66 limited to cilia maintenance/degradation is somewhat weak. Moreover, some experimental conditions are inadequate to verify their hypotheses. Listed below are several main concerns that must be addressed.

Major comments:

1. I do not understand the condition inducing cilia assembly in the IMCD3 cell line utilized in this study. The authors stated that RPE1 cells assemble cilia via the intracellular pathway, whereas IMCD cells assemble cilia via the extracellular pathway. However, they induced ciliogenesis by serum starvation in IMCD as well as in RPE1 cells. Is there any evidence that these two different cell types respond/utilize intracellular or extracellular pathways differently under the same conditions, such as serum starvation?

2. Based on the authors' data, it may be right that Ccdc66 may not be essential for initiating ciliary assembly. However, it is clear that depletion of Ccdc66 in serum-starved IMCD cells results in shorter cilia. Therefore, the data may be interpreted as Ccdc66 deficiency resulting in incomplete ciliary assembly with abnormal ciliary length rather than promoting ciliary disassembly affecting microtubule stability. More concerning is that the authors are considering cells with short cilia as normally ciliated cells. Many studies still report that short cilia are a cilia assembly defect, thus the authors need to clarify the role of Ccdc66 for cilia assembly in IMCD3 cells.

3. In the EV analysis, I do not understand why the authors compared EVs between control cells under serum starvation and Ccdc66-depleted cells under serum addition. To distinguish that EVs from Ccdc66-depleted cells originated from cilia, it would be more feasible for the authors to compare EVs between control and Ccdc66-deficient cells under conditions of ciliary disassembly, i.e., serum addition.

4. In fig.5 experiments, why did the authors compare data from the ciliogenesis proteome of control cells to the ciliary disassembly interactome of Ccdc66-deficient cells? In addition, the authors should compare their data directly with proteomes changed under the same condition, i.e., cilia disassembly condition following serum addition of IMCD, instead of known ciliary disassembly proteome data sets.

5. In this study, the authors mainly have claimed that Ccdc66 has little effect on the initiation of cilia assembly in IMCD cells. However, as shown in Fig.6, in cells depleted of Ccdc66, both Shh and Wnt signaling, which are known to be essential signals for initial cilia assembly, were affected. How do the authors explain this kind of discrepancy about the role of Ccdc66 on the cilia assembly mechanism in IMCD?

Minor comments:

1. Ciliary glycylation has been reported to be involved in regulating cilia length (J Cell Biol. 2017 Sep 4;216(9):2701-2713). Have the authors checked out microtubule glycylation in Ccdc66-depleted cells?

2. In Fig. 6E-G, the authors suggest that the localization of the IFT-B complex during ciliary transport is unaffected by Ccdc6 depletion. However, another report (J Cel Biol 2012 Jun 11;197(6):789-800) indicates that the IFT-A complex is required for Shh signaling. Have the authors checked out the localization of the IFT-A complex in Ccdc66-depleted cells?

3. Where is the data in Figure 2D that corresponds to the sentence "Ccdc66-depleted cells... despite apparent ciliary defects..." on page 8?

4. On page 11, line 8, Fig. 2B should be corrected to Fig. 2C.

5. Microtubule stability is generally measured by the extent of alpha-tubulin acetylation. Are in vitro assays using alpha-tubulin sufficient to demonstrate microtubule stability? Additionally, in the sample treated with 100nM MBP-mNG Ccdc66, alpha-tubulin appears to be present, but the structure appears to be aggregated. Can this be considered a stable state?

6. Images in Fig. S2 show that cilia appear shorter in Flag-mini Turbo-Ccdc66- overexpressed shCtrl cells compared to other figures. For accurate comparison, I suggest showing data from shCtrl cells overexpressing Flag-mini Turbo.

7. On page 16, Fig. S5D should be corrected to Fig. 5D.

8. On page 18, it is stated that "the percentage of SMO-positive cilia also decreased", but there appears to be no significant difference between shCtrl and shCcdc66 in the Fig. 6B graph. The data should be confirmed.

9. In Fig.6C-D, the qRT-PCR data were analyzed using a one-sample t-test, but the error bar range for the shCcdc66 data appear quite wide. Why not analyze the data using other statistics, such as unpaired t-test?

Rev. 2:

The paper by Deretic et al. on "Ccdc66 regulates primary cilia stability, disassembly and signaling important for epithelial organization" aims to demonstrate the role of the microtubule associated protein (MAP) Ccdc66 in the regulation of cilia disassembly through ectocytosis. It also addresses the role of this protein in Hedgehog and WNT signaling and in epithelial organization. As they stand, the data does not fully support these conclusions. The different observations are also not very well linked together.

For all these reasons, I do not think the manuscript is suitable as it stands for publication in Plos Biology.

Major points:

1) Depletion of Ccdc66 in serum starved cells resulted in shorter cilia. A phenotype already described in one of the previous manuscript on Ccdc66 by the same laboratory (Odabasi et al.). The authors should address whether the difference in dynamic observed in this steady state cilia or during disassembly is not a consequence of cilia length (Fig.1 and 4).Similarly, the differences observed in the percentage of ciliated cells following serum addition could be a consequence of short cilia disassembling faster than long cilia (Fig.3) and not a specific role of this protein. As it stands it is thus unclear whether Ccdc66 has a function in ciliary disassembly beyond its role in controlling ciliary length.

2) It would also be important to block ectocytosis to see whether the change in cilia size are indeed a consequence of the observed increase in this process and to assess the role of Aurora Kinase and HDAC6 in this context.

3) Primary-cilium derived extra-cellular vesicles may depend on actin function. Cilia length is also known to depend on actin factors. It will thus be important to address whether Ccdc66 potential interaction with actin actors are involved in its control of ciliary length.

4) Statistics are another major issue in this paper. All experiments have been performed only two times. It is therefore really surprising that the authors can perform a Welch test on the percentage of ciliated cells in each of this two experiments.

5) The result on the Hedgehog pathway have already been published in Odabasi et al on a different cell lines. While the results on the WNT pathway are confusing according to current literature. It is also unclear whether this is a consequence of Ccdc66 role on ciliary length/ Ectocytosis.

6) Finally, while it is clear that CCDC66 depletion as phenotypic consequences on spheroid and epithelium organization, it is unclear whether it depends on its role in ciliary disassembly, ciliary length or mitotic progression.

Rev. 3:

In their manuscript, "Ccdc66 regulates primary cilium stability, disassembly and signaling important for epithelial organization," Deretic et al investigate the function of CCDC66, a microtubule associated protein (MAP) in inner medullary collecting duct (IMCD3) kidney epithelial cells. They demonstrate CCDC66 regulates the stability and disassembly of the microtubule axoneme to maintain cilia length through a mechanism involving AurkA and HDAC6. This moves the field forward by revealing a new role for CCDC66 in cilia and places it in a molecular pathway known to regulate the process. Overall, these experiments are thorough, well-controlled, suitably powered and use appropriate statistical analyses so that the data support this conclusion.

I struggle to state the significance of this work however. The problem it purports to solve is "what does CCDC66 do?" But at the end of the manuscript, I am not sure. CCDC66 does not regulate ciliary assembly in IMCD3 cells as the research group previously showed it does in another cell line, retinal pigment epithelial (RPE) cells. While the authors correctly point out the 2 cells lines use distinct ciliary assembly mechanisms (intracellular vs extracellular), it remains unclear whether this explains the difference. There are certainly growing data to show cell type specific roles for cilia-associated proteins. Perhaps CCDC66 has cell type specific roles then. However, the manuscript barely speculates on this point.

Instead, the manuscript goes on to use proximity labeling "to identify novel proteins that may regulate ciliary length and stability." Again, the work is well-controlled and thorough. The authors perform the PPL at 2 timepoints during disassembly and compare to an existing dataset in the same cell line when ciliated. However, I am unclear what we learn. There is a list of proteins in proximity to CCDC66 under different conditions (or time points). Some make sense from the literature (so by definition are not novel). None are manipulated to determine whether they are, in fact, novel regulators of ciliary length and/or stability. Furthermore, I am not convinced the authors are correct to focus on the 202 proteins that are shared among the ciliated and disassembling cells. Couldn't CCDC6 only come into proximity of proteins regulating disassembly during disassembly? If so, that would make the 82 proteins shared between the disassembly timepoints and not in the ciliated cells of potential interest to prioritize. Regardless, the PPL represents a great deal of work that is an important starting place for understanding the mechanism of CCDC66 in disassembly. I am sure the authors agree performing such work is outside the scope of this work yet without it, we are left with the manuscript revealing a new role of CCDC66 regulating the stability and disassembly of the microtubule axoneme to maintain cilia length and a long list of proteins that are near CCDC66.

Finally, the manuscript examines some functional consequences of CCDC66 depletion. It argues that CCDC66 depletion results in dampened Hh and Wnt response. In the Hh work, the authors need to show the Gli1 and Ptch1 levels in unstimulated as well as stimulated conditions. They use SAG which directly binds and activates SMO to stimulate yet also show a defect in SMO ciliary enrichment. This is confusing. First, I acknowledge that ciliary SMO enrichment does not correlate to pathway activation. But if CCDC66 is regulating that enrichment, it appears that CCDC66 acts upstream of SMO activation. If so, then SAG stimulation should overcome that function. Are the data arguing that CCDC66 regulated Hh at a step upstream of SMO and an additional step downstream of activated SMO? The authors could try the new phosphorylated SMO antibody characterized in PMID: 37214942 as it is commercially available and could at least get at whether CCDC66 regulates the activation of SMO. In the Wnt work, the authors need to acknowledge the everchanging and controversial relationship of cilia and Wnt signaling. And they need to provide a much clearer, data-based framework from which to interpret these experiments. Are IMCD3 cells Wnt responsive? What Wnt ligand? What are the transcriptional targets (Axin2 seems highly likely but are the others know in IMCD3 cells)? And again, the authors need to show the Axin2, Fibronectin and Pdk1 transcript levels in unstimulated as well as stimulated conditions.

Minor points:

Supp Fig 1 shows specificity of Ab, and also shows heterogeneity in KD by siRNA, which is to be expected for the technique. This is clear in MB staining (which I assume is midbody) where CCDC66 staining persistence is visible in top right of green inset. Additionally, there is green signal overlapping the acTub staining on the left side of the shCcdc66 panel that looks ciliary. Broader point is that it appears that the depletion overall is strong but varies cell by cell making it likely that much of the variability in phenotype may be driven by the KD technique. This should be mentioned. At one level, that further supports the conclusion that CCDC66 regulates the stability and disassembly of the microtubule axoneme - but in terms of signaling and broader functional consequences, the variability make the data challenging to interpret.

In methods, description of In vitro microtubule stabilization assay is confusing. It sounds like there was no GTP added to buffers with the indicated proteins (100 nM of protein His-MBP-mNeonGreen, His-MBP- mNeonGreen-Ccdc66 or His-MBP-TOGARAM1TOG1TOG2). The text of the manuscript suggests there is GTP. This needs to be clarified for the experiment to be interpretable.

The authors should specify the full timeline of the various manipulations. They state that the shRNA treatment requires 6 days. They also serum starve and serum starve then serum stimulate. Is this all sequential? In other words, are there instances when the serum starvation starts prior to the 6 days of shRNA treatment? Or other overlap. It needs to be clear for others to repeat.

In the abstract, I believe the authors broad point, that cilia disassembly is less well understood than cilia assembly is true. However, many would take issue with the statement that "cilium assembly is well-understood", most notably those who study it and possess many questions. They deserve a modest rephrasing.

Overall, the manuscript brings in a great deal of the literature yet there are quite a few instances where references are missing. This will simply require a rigorous proofread.

On page 17 of the manuscript, the authors state, "Although inhibiting AurkA and HDAC6 rescued Ccdc66 disassembly phenotypes, our analysis did not detect these key players in cilium disassembly interactome. This indicates that Ccdc66 has an indirect relationship with these proteins." The final statement is inaccurate as PPL not picking up a protein only means that it did not pick up that protein. There are many reasons direct interactors may not be detected at specific times or under specific conditions using PPL-based methods.

It would be far easier to follow when the authors are discussing protein or transcript if the authors adhered to nomenclature guidelines of mammalian proteins being in capital letters and genes being in italics (and caps in the case of human).

---

## [Decision Letter · Decision Letter 2]

Dear Elif,

Thank you for your patience while we considered your revised manuscript entitled "CCDC66 regulation of cytoskeleton and cilia stability is important for signaling and epithelial organization" for consideration as a Research Article at PLOS Biology. Your revised study has now been evaluated by the PLOS Biology editors, the Academic Editor and the three original reviewers.

The reviews are attached below. You will see that the reviewers appreciate all the work you have done in the revision and are mostly satisfied with the responses and additional analyses. However, Reviewer 2 has raised several points that should be clarified before we can accept the manuscript for publication, including that some of the experiments were done on assembling/steady state cilia, whereas others were done on disassembling cilia and that this makes the paper difficult to follow and would be better to show both processes in the same figures for comparison. This reviewer also notes a few points that are seemly contradictory and should be explained further in the text and some missing controls. Reviewer 3 remains unconvinced that the findings are sufficiently significant for PLOS Biology. Although we appreciate these comments, we agree with the other two reviewers and have decided to proceed and invite a final round of revision.

In light of the reviews, which you will find at the end of this email, we are pleased to offer you the opportunity to address the remaining points from Reviewer 2 in a revision that we anticipate should not take you very long. We will then assess your revised manuscript and your response to the reviewers' comments with our Academic Editor aiming to avoid further rounds of peer-review, although we might need to consult with the reviewers, depending on the nature of the revisions.

In addition to these revisions, you will need to complete some formatting changes, which you will receive in a follow up email. A member of our team will be in touch with a set of requests shortly. We expect to receive your revised manuscript within 1 month. Please email us (plosbiology@plos.org) if you have any questions or concerns, or would like to request an extension.

**IMPORTANT - SUBMITTING YOUR REVISION**

3. Resubmission Checklist

Sincerely,

Ines

--

Ines Alvarez-Garcia, PhD

Senior Editor

PLOS Biology

Reviewers' comments

Rev. 1: Ji Eun Lee - please note that this reviewer has signed the review

I have thoroughly reviewed the authors' rebuttal letter, revised manuscript text, figures, and supplementary materials. The authors have comprehensively and convincingly addressed my comments, Specifically:

1. The authors provided experimental evidence differentiating roles of CCDC66 in ciliogenesis, axoneme stabilization, and disassembly processes, addressing my initial concerns.

2. Additional data regarding ectocytosis and the involvement of the actin cytoskeleton and microtubule stabilization provided substantial mechanistic clarity.

3. The statistical robustness of the data has improved, fulfilling all requested criteria.

4. Revised explanations and additional data on the Hedgehog and Wnt signaling pathways further clarified the manuscript's conclusions.

5. The graphical model and explicit discussion added by the authors now effectively convey the integration of their findings and potential physiological relevance.

Overall, the revisions have substantially strengthened the manuscript, resolving all the concerns raised in the previous round of review.

Therefore, I am pleased to recommend acceptance of this manuscript for publication in PLOS Biology.

Rev. 2:

The paper by Deretic et al. on "CCDC66 regulation of cytoskeleton and cilia stability is important for signaling and epithelial organization" has been really improved by the major revisions done. The authors respond to all my previous comments. However, the substantial changes done to the paper raised a few new issues that should be addressed prior to publication.

Notably, some experiments were done on assembling/steady state cilia while others were done on disassembling cilia. While it might be that both processes use the same mechanisms to control ciliary length, it makes the paper difficult to follow. Putting both processes on the same figures to compare (including in the model) may help. There is also seemingly contradictory results that make the understanding very hard and are not discussed.

My understanding of the authors hypothesis is that during cilium assembly and maintenance, primary cilia underwent few events of ectocytosis (mainly EVs) while microtubules are stable. Events of ectocytosis may dependent on the actin cytoskeleton. In absence of CCDC66, ectocytosis is increased and microtubule destabilized leading to their fragmentation. Thus, at this stage like during disassembly, cytoD should at least partially rescue ciliary length and decrease ectocytosis both in control and Ccdc66 depleted cells. Taxol should lead to primary cilia of the same size in control and depleted cells, maybe slightly smaller than in none treated cells due to the decrease availability of free tubulin. Could the authors check whether it works according to their model?

Intriguingly, CCDC66 may interact and bundle actin (difficult to say without the control image of the actin network in vitro in figure S5A). Accordingly, its depletion seems to decrease cytoplasmic actin cables. Therefore, we would expect less actin in the primary cilia and thus less ectocytosis in depleted cells, the reverse of what is observed. How do the authors reconcile these paradoxical results? Could it be that depletion of Ccdc66 leads to a decrease accumulation of actin at the tip and more shedding events all along the primary cilia?

During primary cilia disassembly, disassembly and cilia shortening depends on the actin cytoskeleton. However, ectocytosis seems not to be affected in control by Cytochalasin D treatment. Thus, what could be the role of actin? Disassembly but not cilia shortening (isn't it paradoxical?) also depends on microtubule stability through HDAC6/Aurora A. In absence of CCDC66, there is more events of cilia fragmentation that depends at least partially of the actin cytoskeleton and maybe of microtubule destabilization.

Minor points:

* There are a few inaccuracies that should be removed:

- Primary cilia are not only assembled in cell exiting the cell cycle, but observed in most cells in G1.

- A cell entering mitosis cannot be quiescent.

- On page 7 the results on Cep164, Talpid3 and CP110 are described two times.

- It is not always clear whether statistics were applied on the three repeats of an experiment or on all the values measured in these three experiments. Please clearify.

- There is still experiments with only two repeats in the supplementary data.

- There is some inaccuracies on the figure numbering in the text.

- Cytochalasin D affects both branched and linear actin filaments.

- It would help to mention in the figure legend against what were normalized quantifications.

- There is a problem with the legend of at least Figure 6 and S5

* What means intact cilia?

* Over interpretations should be avoided. E.g., The paper do not demonstrate that CCDC66 plays a role in activating HH and Wnt by regulating intraciliary and vesicular transport.

Rev. 3:

In their manuscript, "CCDC66 regulation of cytoskeleton and cilia stability is important for signaling and epithelial organization," Deretic et al investigate the role of CCDC66 using IMCD3 cells. This is a revised manuscript for which the authors have completed a great deal of additional work including revisions to the writing and through additional experiments. At the heart of my previous review is the issue of significance and I still struggle to see the significance of the work- meaning why the results are important. The authors offer that they ascribe function to CCDC66 in terms of stabilizing microtubules. Indeed, the authors themselves previously published this finding (PLoS Biol. 20, e3001708) and discussed it in a previous review (J Cell Science. 136 (23); others have also reported it. I understand this manuscript provides additional evidence but the significance is incremental.

The revision does individually address each of the many reviewer comments, and the laundry list of experimental observations grows but, at the end, I don't have a better understanding of why these results are important. As the PLoS Biology mission is specifically, "to publish significant advances across all areas of biological science", the manuscript does not reach the necessary bar.

---

## [Editor Report · Decision Letter 3]

Dear Elif,

Thank you for your patience while we considered your revised manuscript entitled "CCDC66 regulation of cytoskeleton and cilia stability is important for signaling and epithelial organization" for publication as a Research Article at PLOS Biology. This revised version of your manuscript has been evaluated by the PLOS Biology editors and by the Academic Editor.

Based on our Academic Editor's assessment of your revision, we are likely to accept this manuscript for publication, provided you satisfactorily address the data and other policy-related requests stated below my signature.

We expect to receive your revised manuscript within two weeks.

*Published Peer Review History*

*Press*

Sincerely,

Ines

--

Ines Alvarez-Garcia, PhD

Senior Editor

PLOS Biology

DATA POLICY:

Fig. 1C, D, F, G; Fig. 2B, E; Fig. 3B, E, F; Fig. 4A-D; Fig. 6C, E, F, H; Fig. 7B-G; Fig. 8B-D, F, G; Fig. S1C-F; Fig. S2B, C, E, F; Fig. S3A-C; Fig. S4C-F; Fig. S5D, E and Fig. S6B, D, F, H, I, L

---

## [Editor Report · Decision Letter 4]

Dear Elif,

Thank you for the submission of your revised Research Article entitled "CCDC66 regulation of cytoskeleton and cilia stability is important for signaling and epithelial organization" for publication in PLOS Biology. On behalf of my colleagues and the Academic Editor, Renata Basto, I am delighted to let you know that we can in principle accept your manuscript for publication, provided you address any remaining formatting and reporting issues. These will be detailed in an email you should receive within 2-3 business days from our colleagues in the journal operations team; no action is required from you until then. Please note that we will not be able to formally accept your manuscript and schedule it for publication until you have completed any requested changes.

PRESS

Sincerely, 

Ines

--

Ines Alvarez-Garcia, PhD

Senior Editor

PLOS Biology
